# Disentangling the Predictive Variance of Deep Ensembles through the Neural Tangent Kernel

**Seijin Kobayashi**
Department of Computer Science
ETH Zürich
seijink@ethz.ch

**Pau Vilimelis Aceituno**
Institute of Neuroinformatics
University of Zürich & ETH Zürich
pau@ini.ethz.ch

**Johannes von Oswald**
Department of Computer Science
ETH Zürich
voswaldj@ethz.ch

## Abstract

Identifying unfamiliar inputs, also known as out-of-distribution (OOD) detection, is a crucial property of any decision making process. A simple and empirically validated technique is based on deep ensembles where the variance of predictions over different neural networks acts as a substitute for input uncertainty. Nevertheless, a theoretical understanding of the inductive biases leading to the performance of deep ensemble's uncertainty estimation is missing. To improve our description of their behavior, we study deep ensembles with large layer widths operating in simplified linear training regimes, in which the functions trained with gradient descent can be described by the neural tangent kernel. We identify two sources of noise, each inducing a distinct inductive bias in the predictive variance at initialization. We further show theoretically and empirically that both noise sources affect the predictive variance of non-linear deep ensembles in toy models and realistic settings after training. Finally, we propose practical ways to eliminate part of these noise sources leading to significant changes and improved OOD detection in trained deep ensembles.

## 1 Introduction

Modern artificial intelligence uses intricate deep neural networks to process data, make predictions and take actions. One of the crucial steps toward allowing these agents to act in the real world is to incorporate a reliable mechanism for estimating uncertainty – in particular when human lives are at risk [1, 2]. Although the ongoing success of deep learning is remarkable, the increasing data, model and training algorithm complexity make a thorough understanding of their inner workings increasingly difficult. This applies when trying to understand when and why a system is certain or uncertain about a given output and is therefore the topic of numerous publications [3–10].

Principled mechanisms for uncertainty quantification would rely on Bayesian inference with an appropriate prior. This has led to the development of (approximate) Bayesian inference methods for deep neural networks [11–15]. Simply aggregating an ensemble of models [16] and using the disagreement of their predictions as a substitute for uncertainty has gained popularity. However, the theoretical justification of deep ensembles remains a matter of debate, see Wilson and Izmailov [17]. Although a link between Bayesian inference and deep ensembles can be obtained, see [18, 19], an understanding of the widely adopted *standard* deep ensemble and it's predictive distribution is still missing [20, 21]. Note that even for principled Bayesian approaches there is no valid theoretical or practical OOD guarantee without a proper definition of out-of-distribution data [22].

36th Conference on Neural Information Processing Systems (NeurIPS 2022).

One avenue to simplify the analyses of deep neural networks that gained a lot of attention in recent years is to increase the layer width to infinity [23, 24] or to very large values [25, 26]. In the former regime, an intriguing equivalence of infinitely wide deep networks at initialization and Gaussian processes allows for exact Bayesian inference and therefore principled uncertainty estimation. Although it is not possible to generally derive a Bayesian posterior for *trained* infinite or finite layer width networks, the resulting model predictions can be expressed analytically by kernels. Given this favorable mathematical description, the question of how powerful and similar these models are compared to their arguably black-box counterparts arises, with e.g. moderate width, complex optimizers and training stochasticity [25, 27–32].

In this paper, we leverage this tractable description of trained neural networks and take a first step towards understanding the predictive distribution of neural networks ensembles with large but finite width. Building on top of the various studies mentioned, we do so by studying the case where these networks can be described by a kernel and study the effect of two distinct noise sources stemming from the network initialization: The noise in the functional initialization of the network and the initialization noise of the gradient, which affects the training and therefore the kernel. As we will show, these noise sources will affect the predictive distributions differently and influence the network's generalization on in- and out-of-distribution data.

Our contributions are the following:

- We provide a first order approximation of the predictive variance of an ensemble of linearly trained, finite-width neural networks. We identify interpretable terms in the refined variance description, originating from 2 distinct noise sources, and further provide their analytical expression for single layer neural networks with ReLU non-linearities.

- We show theoretically that under mild assumptions these refined variance terms survive nonlinear training for sufficiently large width, and therefore contribute to the predictive variance of non-linearly trained deep ensembles. Crucially, our result suggests that any finer description of the predictive variance of a linearized ensemble can be erased by nonlinear training.

- We conduct empirical studies validating our theoretical results, and investigate how the different variance terms influence generalization on in - and out-of-distribution. We highlight the practical implications of our theory by proposing simple methods to isolate noise sources in realistic settings which can lead to improved OOD detection.[1]

## 2  Neural network ensembles and their relations to kernels

Let $f_\theta = f(\cdot, \theta) : \mathbb{R}^{h_0} \to \mathbb{R}^{h_L}$ denote a neural network parameterized by the weights $\theta \in \mathbb{R}^n$. The weights consist of weight matrices and bias vectors $\{(W_l, b_l)\}_{l=1}^L$ describing the following feed-forward computation beginning with the input data $x^0$:

$$z^{l+1} = \frac{\sigma_w}{\sqrt{h_l}} W^{l+1} x^l + b^{l+1} \text{ with } x^{l+1} = \phi(z^{l+1}). \tag{1}$$

Here $h_l$ is the dimension of the vector $x^l$ and $\phi$ is a pointwise non-linearity such as the softplus $\log(1 + e^x)$ or Rectified Linear Unit i.e. $\max(0, x)$ (ReLU) [33]. We follow Jacot et al. [24] and use $\sigma_w = \sqrt{2}$ to control the standard deviation of the initialised weights $W_{ij}^l, b_i^l \sim \mathcal{N}(0, 1)$.

Given a set of $N$ datapoints $\mathcal{X} = (x_i)_{0 \le i \le N} \in \mathbb{R}^{N \times h_0}$ and targets $\mathcal{Y} = (y_i)_{0 \le i \le N} \in \mathbb{R}^{N \times h_L}$, we consider regression problems with the goal of finding $\theta^*$ which minimizes the mean squared error (MSE) loss $\mathcal{L}(\theta) = \frac{1}{2} \sum_{i=0}^N \|f(x_i, \theta) - y_i\|_2^2$. For ease of notation, we denote by $f(\mathcal{X}, \theta) \in \mathbb{R}^{N \cdot h_L}$ the vectorized evaluation of $f$ on each datapoint and $\mathcal{Y} \in \mathbb{R}^{N \cdot h_L}$ the target vector for the entire dataset. As the widths of the hidden layers grow towards infinity, the distribution of outputs at initialization $f(x, \theta_0)$ converges to a multivariate gaussian distribution due to the Central Limit Theorem [23]. The resulting function can then accurately be described as a zero-mean Gaussian process, coined Neural Network Gaussian Process (NNGP), where the covariance of a pair of output neurons $i, j$ for data $x$ and $x'$ is given by the kernel

---

[1]Source code for all experiments: `github.com/seijin-kobayashi/disentangle-predvar`

$$\mathcal{K}(x,x')^{i,j} = \lim_{h\to\infty} \mathbb{E}[f^i(x,\theta_0)f^j(x',\theta_0)] \tag{2}$$

with $h = \min(h_1, ..., h_{L-1})$. This equivalence can be used to analytically compute the Bayesian posterior of infinitely wide Bayesian neural networks [34].

On the other hand infinite width models *trained* via gradient descent (GD) can be described by the Neural Tangent Kernel (NTK). Given $\theta$, the NTK $\Theta_\theta$ of $f_\theta$ is a matrix in $\mathbb{R}^{N\cdot h_L} \times \mathbb{R}^{N\cdot h_L}$ with the $(i,j)$-entry given as the following dot product

$$\langle \nabla_\theta f(x_i,\theta), \nabla_\theta f(x_j,\theta)\rangle \tag{3}$$

where we consider without loss of generality the output dimension of $f$ to be $h_L = 1$ for ease of notation. Furthermore, we denote $\Theta_\theta(\mathcal{X},\mathcal{X}) := \nabla_\theta f(\mathcal{X},\theta)\nabla_\theta f(\mathcal{X},\theta)^T$ the matrix and $\Theta_\theta(x',\mathcal{X}) := \nabla_\theta f(x',\theta)\nabla_\theta f(\mathcal{X},\theta)^T$ the vector form of the NTK while highlighting the dependencies on different datapoints.

Lee et al. [25] showed that for sufficiently wide networks under common parametrizations, the gradient descent dynamics of the model with a sufficiently small learning rate behaves closely to its linearly trained counterpart, i.e. its first-order Taylor expansion in parameter space. In this gradient flow regime, after training on the mean squared error converges, we can rewrite the predictions of the linearly trained models in the following closed-form:

$$f^{\text{lin}}(x) = f(x,\theta_0) + \mathcal{Q}_{\theta_0}(x,\mathcal{X})(\mathcal{Y} - f(\mathcal{X},\theta_0)) \tag{4}$$

where $\mathcal{Q}_{\theta_0}(x,\mathcal{X}) := \Theta_{\theta_0}(x,\mathcal{X})\Theta_{\theta_0}(\mathcal{X},\mathcal{X})^{-1}$ with $\Theta_{\theta_0}$ the NTK at initialization, i.e. of $f(.,\theta_0)$. The linearization error throughout training $\sup_{t\geq 0}\|f_t^{\text{lin}}(x) - f_t(x)\|$ is further shown to decrease with the width of the network, bounded by $\mathcal{O}(h^{-\frac{1}{2}})$. Note that one can also linearize the dynamics without increasing the width of a neural network but by simply changing its output scaling [26].

When moving from finite to the infinite width limit the training of a multilayer perceptron (MLP) can again be described with the NTK, which now converges to a deterministic kernel $\Theta_\infty$ [24], a result which extends to convolutional neural networks [27] and other common architectures [35, 36]. A fully trained neural network model can then be expressed as

$$f_\infty(x) = f(x,\theta_0) + \Theta_\infty(x,\mathcal{X})\Theta_\infty(\mathcal{X},\mathcal{X})^{-1}(\mathcal{Y} - f(\mathcal{X},\theta_0)). \tag{5}$$

where $f(\{\mathcal{X},x\},\theta_0) \sim \mathcal{N}(0,\mathcal{K}(\{\mathcal{X},x\},\{\mathcal{X},x\}))$.

## 2.1 Predictive distribution of *linearly trained* deep ensembles

In this Section, we study in detail the predictive distribution of ensembles of linearly trained models, i.e. the distribution of $f^{lin}(x)$ given $x$ over random initializations $\theta_0$. In particular, for a given data $x$, we are interested in the mean $\mathbb{E}[f(x)]$ and variance $\mathbb{V}[f(x)]$ of trained models over random initialization. The former is typically used for the prediction of a deep ensemble, while the latter is used for estimating model or epistemic uncertainty utilized e.g. for OOD detection or exploration. To start, we describe the simpler case of the infinite width limit and a deterministic NTK, which allows us to compute the mean and variance of the solutions found by training easily:

$$\begin{aligned}
\mathbb{E}[f_\infty(x)] &= \mathcal{Q}_\infty(x,\mathcal{X})\mathcal{Y}, \\
\mathbb{V}[f_\infty(x)] &= \mathcal{K}(x,x) + \mathcal{Q}_\infty(x,\mathcal{X})\mathcal{K}(\mathcal{X},\mathcal{X})\mathcal{Q}_\infty(x,\mathcal{X})^T - 2\mathcal{Q}_\infty(x,\mathcal{X})\mathcal{K}(\mathcal{X},x)
\end{aligned} \tag{6}$$

where we introduced $\mathcal{Q}_\infty(x,\mathcal{X}) = \Theta_\infty(x,\mathcal{X})\Theta_\infty(\mathcal{X},\mathcal{X})^{-1}$.

For finite width linearly trained networks, the kernel is no longer deterministic, and its stochasticity influences the predictive distribution. Because there is probability mass assigned to the neighborhood of rare events where the NTK kernel matrix is not invertible, the expectation and variance over parameter initialization of the expression in equation 4 diverges to infinity.

Fortunately, due to the convergence in probability of the empirical NTK to the infinite width counterpart [24], we know these singularities become rarer and ultimately vanish as the width increases to infinity. Intuitively, we should therefore be able to assign meaningful, finite values to these undefined

quantities, which ignores these rare singularities. The delta method [37] in statistics formalizes this intuition, by using Taylor approximation to smooth out the singularities before computing the mean or variance. When the probability mass of the empirical NTK is highly concentrated in a small radius around the limiting NTK, the expression 4 is roughly linear w.r.t the NTK entries. Given this observation, we prove (see Appendix A.2) the following result, and justify that the obtained expression is informative of the *empirical* predictive mean and variance of deep ensembles. Rewriting equation 4 into

$$
\begin{aligned}
f^{\text{lin}}(x) =& f(x, \theta_0) + \bar{\mathcal{Q}}(x, \mathcal{X})(\mathcal{Y} - f(\mathcal{X}, \theta_0)) \\
& + [\mathcal{Q}_{\theta_0}(x, \mathcal{X}) - \bar{\mathcal{Q}}(x, \mathcal{X})](\mathcal{Y} - f(\mathcal{X}, \theta_0))
\end{aligned}
\tag{7}
$$

where $\bar{\mathcal{Q}}(x, \mathcal{X}) = \bar{\Theta}(x, \mathcal{X})\bar{\Theta}(\mathcal{X}, \mathcal{X})^{-1}$ and $\bar{\Theta} = \mathbb{E}(\Theta_{\theta_0})$, we state:

**Proposition 2.1.** *For one hidden layer networks parametrized as in equation 1, given an input $x$ and training data $(\mathcal{X}, \mathcal{Y})$, when increasing the hidden layer width $h$, we have the following convergence in distribution over random initialization $\theta_0$:*

$$
\sqrt{h}[\mathcal{Q}_{\theta_0}(x, \mathcal{X}) - \bar{\mathcal{Q}}(x, \mathcal{X})](\mathcal{Y} - f(\mathcal{X}, \theta_0)) \overset{dist.}{\to} Z(x)
$$

*where Z(x) is the linear combination of 2 Chi-Square distributions, such that*

$$
\mathbb{V}(Z(x)) = \lim_{h \to \infty} (h\mathbb{V}^c(x) + h\mathbb{V}^i(x))
$$

*where*

$$
\begin{aligned}
\mathbb{V}^c(x) =& \mathbb{V}[\Theta_{\theta_0}(x, \mathcal{X})\bar{\Theta}(\mathcal{X}, \mathcal{X})^{-1}\mathcal{Y}] + \mathbb{V}[\bar{\mathcal{Q}}(x, \mathcal{X})\Theta_{\theta_0}(\mathcal{X}, \mathcal{X})\bar{\Theta}(\mathcal{X}, \mathcal{X})^{-1}\mathcal{Y}] \\
& - 2\mathbb{C}ov[\bar{\mathcal{Q}}(x, \mathcal{X})\Theta_{\theta_0}(\mathcal{X}, \mathcal{X})\bar{\Theta}(\mathcal{X}, \mathcal{X})^{-1}\mathcal{Y}, \Theta_{\theta_0}(x, \mathcal{X})\bar{\Theta}(\mathcal{X}, \mathcal{X})^{-1}\mathcal{Y}], \\
\mathbb{V}^i(x) =& \mathbb{V}[\Theta_{\theta_0}(x, \mathcal{X})\bar{\Theta}(\mathcal{X}, \mathcal{X})^{-1}f(\mathcal{X}, \theta_0)] + \mathbb{V}[\bar{\mathcal{Q}}(x, \mathcal{X})\Theta_{\theta_0}(\mathcal{X}, \mathcal{X})\bar{\Theta}(\mathcal{X}, \mathcal{X})^{-1}f(\mathcal{X}, \theta_0)] \\
& - 2\mathbb{C}ov[\bar{\mathcal{Q}}(x, \mathcal{X})\Theta_{\theta_0}(\mathcal{X}, \mathcal{X})\bar{\Theta}(\mathcal{X}, \mathcal{X})^{-1}f(\mathcal{X}, \theta_0), \Theta_{\theta_0}(x, \mathcal{X})\bar{\Theta}(\mathcal{X}, \mathcal{X})^{-1}f(\mathcal{X}, \theta_0)].
\end{aligned}
$$

We omit the dependence of $\theta_0$ on the width $h$ for notational simplicity. While the expectation or variance of equation 4 for any finite width is undefined, their empirical mean and variance are with high probability indistinguishable from that of the above limiting distribution (see Lemma A.1). Note that the above proposition assumes the noise in $\Theta_{\theta_0}$ to be decorrelated from $f(x, \theta_0)$, which can hold true under specific constructions of the network that are of practical interest as we will see in the following (c.f. Appendix A.3.2).

Given Proposition 2.1, we now describe the approximate variance of $f^{\text{lin}}(x)$ for $L = 2$, which we can extend to the general $L > 2$ case using an informal argument (see A.2.2):

**Proposition 2.2.** *Let $f$ be a neural network with identical width of all hidden layers, $h_1 = h_2 = ... = h_{L-1} = h$. We assume $\|\Theta_{\theta_0} - \bar{\Theta}\|_F^2 = \mathcal{O}_p(\frac{1}{h})$. Then,*

$$
\mathbb{V}[f^{lin}(x)] \approx \mathbb{V}^a(x) + \mathbb{V}^c(x) + \mathbb{V}^i(x) + \mathbb{V}^{cor}(x) + \mathbb{V}^{res}(x)
$$

*where*

$$
\begin{aligned}
\mathbb{V}^a(x) =& \bar{\mathcal{K}}(x, x) + \bar{\mathcal{Q}}(x, \mathcal{X})\bar{\mathcal{K}}(\mathcal{X}, \mathcal{X})\bar{\mathcal{Q}}(x, \mathcal{X})^T - 2\bar{\mathcal{Q}}(x, \mathcal{X})\bar{\mathcal{K}}(\mathcal{X}, x), \\
\mathbb{V}^{cor}(x) =& 2\mathbb{E}\big[[\Theta_{\theta_0}(x, \mathcal{X}) - \bar{\mathcal{Q}}(x, \mathcal{X})\Theta_{\theta_0}(\mathcal{X}, \mathcal{X})][\bar{\Theta}(\mathcal{X}, \mathcal{X})^{-1}\Theta_{\theta_0}(\mathcal{X}, \mathcal{X})\bar{\Theta}(\mathcal{X}, \mathcal{X})^{-1}]\big] \\
& \cdot [\bar{\mathcal{K}}(\mathcal{X}, x) - \bar{\mathcal{K}}(\mathcal{X}, \mathcal{X})\bar{\mathcal{Q}}(x, \mathcal{X})^T]
\end{aligned}
$$

*and $\mathbb{V}^{res}(x) = \mathcal{O}(h^{-2})$ as well as $\bar{\mathcal{K}}$ the expectation over initializations of the finite width counterpart of the NNGP kernel.*

Several observations can be made: First, the above expression only involves the first and second moments of the empirical, finite width NTK, as well as the first moment of the NNGP kernel. These terms can be analytically computed in some settings. We provide in Appendix A.4.3 some of the moments for the special case of a 1-hidden layer ReLU network, and show the analytical expression correspond to empirical findings.

Second, the decomposition demonstrates the interplay of 2 distinct noise sources in the predictive variance:

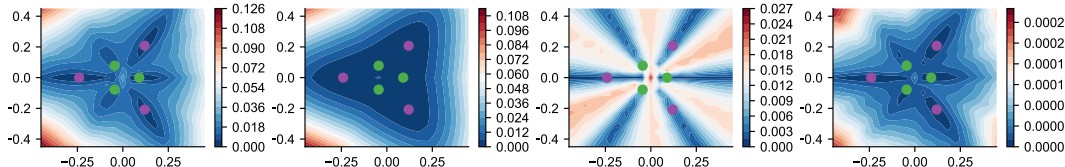

Figure 1: Empirical variance $\mathbb{V}[f(x)]$ of our kernel models $f^{\text{lin}}$, $f^{\text{lin-a}}$, $f^{\text{lin-c}}$ and $f^{\text{lin-i}}$ on a toy regression problem, consisting in regressing against a value of 1 on the purple and -1 on the green datapoints. *Left:* $\mathbb{V}[f^{\text{lin}}(x)]$ as a superposition of the three isolated components right of it. *Center left:* $\mathbb{V}[f^{\text{lin-a}}(x)]$ which correlates with the distance to the datapoints. *Center right:* $\mathbb{V}[f^{\text{lin-c}}(x)]$ which correlates with the angle to the datapoints. *Right:* $\mathbb{V}[f^{\text{lin-i}}(x)]$ which depends on distance and angle to the datapoints.

- $\mathbb{V}^a$ is the variance associated to the expression in the first line of equation 7. Intuitively, it is the finite width counterpart of the predictive variance of the infinite width model (equation 6), as it assumes the NTK is deterministic. The variance stems entirely from the functional noise at initialization and converges to the infinite width predictive variance as the width increases.

- $\mathbb{V}^c$ and $\mathbb{V}^i$ stem from the second line of equation 7. $\mathbb{V}^c$ is a first-order approximation of the predictive variance of a linearly trained network with *pure kernel noise*, without functional noise i.e. $\mathbb{V}^c \approx \mathbb{V}[\mathcal{Q}_{\theta_0}(x, \mathcal{X})\mathcal{Y}]$. On the other hand, $\mathbb{V}^i$ depends on the interplay between the 2 noises, and can be identified as the predictive variance of a deep ensemble with a deterministic NTK $\bar{\Theta}$ and a new functional prior $g(x) = \Theta_{\theta_0}(x, \mathcal{X})\bar{\Theta}(\mathcal{X}, \mathcal{X})^{-1}f(\mathcal{X}, \theta_0)$. Intuitively, this new functional prior can be seen as a data-specific inductive bias on the NTK formulation of the predictive variance (see Appendix A.3.1 for more details).

- $\mathbb{V}^{cor}$ is a covariance term between the 2 terms in equation 7 and also contains the correlation terms between $\Theta_{\theta_0}$ and $f(x, \theta_0)$. In general, its analytical expression is challenging to obtain as it requires the 4th moments of the finite width NNGP kernel fluctuation. Here, we provide its expression under the same simplifying assumption that the noise in $\Theta_{\theta_0}$ is decorrelated from $f(x, \theta_0)$. We therefore do not attempt to describe it in general, and focus in our empirical Section on the terms that are tractable and can be easily isolated for practical purposes.

Each of $\mathbb{V}^c, \mathbb{V}^i, \mathbb{V}^{cor}$ decay in $\mathcal{O}(h^{-1})$, which, together with $\mathbb{V}^a$, provide a first-order approximation of the predictive variance of $f^{lin}(x)$. Note that $\mathbb{V}^a$ and $\mathbb{V}^c$ are of particular interest, as removing either the kernel or the functional noise at initialization will collapse the predictive variance of the trained ensemble to either one of these 2 terms.

## 2.2 Predictive distribution of *standard* deep ensemble of large width

An important question at this point is to which extent our analysis for linearly trained models applies to a fully and non-linearly trained deep ensemble. Indeed, if the discrepancy between the predictive variance of a linearly trained ensemble and its non-linear counterpart is of a larger order of magnitude than the higher-order correction in the variance term, the latter can be 'erased' by training. Building on top of previous work, we show that, under the assumption of an empirically supported conjecture [38], for one hidden layer networks trained on the Mean Squared Error (MSE) loss, this discrepancy is asymptotically dominated by the refined predictive variance terms of the linearly trained ensemble we described in Section 2.1.

**Proposition 2.3.** *Let $f$ be a neural network with identical width of all hidden layers, $h_1 = h_2 = ... = h_{L-1} = h$, and such that the derivative of the non-linearity $\phi'$ is bounded and Lipschitz continuous on $\mathbb{R}$. Let the training data $(\mathcal{X}, \mathcal{Y})$ contained in some compact set, such that the NTK of $f$ on $\mathcal{X}$ is invertible. Let $f_t$ (resp. $f_t^{lin}$) be the model (resp. linearized model) trained on the MSE loss with gradient flow at timestep $t$ with some learning rate. Assuming*

$$\sup_t \|\Theta_{\theta_0} - \Theta_{\theta_t}\|_F = \mathcal{O}(\frac{1}{h}) \tag{8}$$

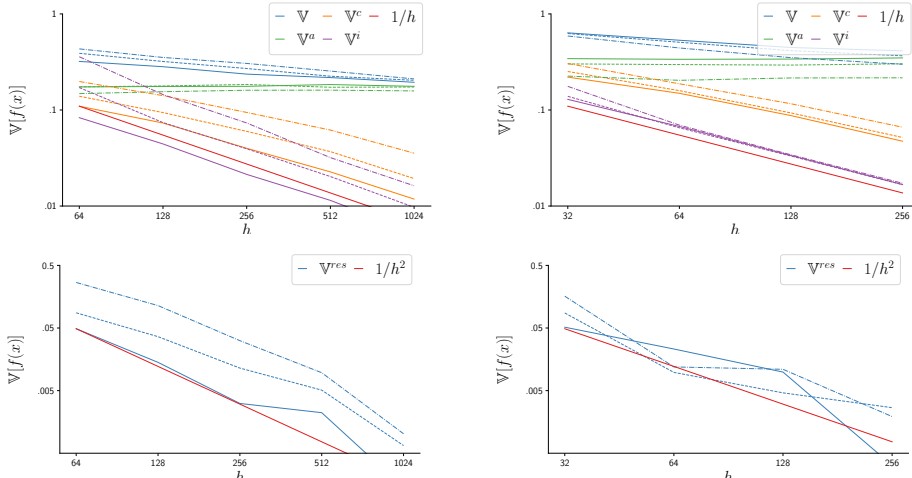

Figure 2: Empirical predictive variance $\mathbb{V}, \mathbb{V}^a, \mathbb{V}^i, \mathbb{V}^c$ of an ensemble of 100 models. *Upper Left:* Linearized feed forward networks of various depths and widths trained on a subset of MNIST. *Upper Right:* Linearized convolutional neural networks of various depth and widths trained on a subset of CIFAR10. *Lower Left (resp. Right):* $1/h^2$ scaling of $\mathbb{V}^{res}$ of an ensemble of MLPs (resp. CNN) of various depths and widths trained on a subset of MNIST (resp. CIFAR10). Although the theoretical result on the scaling of $1/h$ of the variance terms influenced by the kernel noise as well as $1/h^2$ of the residual holds only for depth $L = 2$ (line plots), the same scaling is observed for deeper networks as suggested by our informal result ($L = 3$ in dashed lines, $L = 5$ in dashed-dotted lines). All plots are plotted in the log-log scale.

*Then, $\forall x, \forall \delta > 0, \exists C, H : \forall h > H,$*

$$\mathbb{P}\big[\sup_t \|f_t^{lin}(x) - f_t(x)\|_2 \le \frac{C}{h}\big] \ge 1 - \delta. \tag{9}$$

*In particular, for one hidden layer networks, after training,*

$$|\hat{\mathbb{V}}(f(x)) - \hat{\mathbb{V}}(f^{lin}(x))| = \mathcal{O}_p(\hat{\mathbb{V}}\big[[\mathcal{Q}_{\theta_0}(x, \mathcal{X}) - \bar{\mathcal{Q}}(x, \mathcal{X})](\mathcal{Y} - f(\mathcal{X}, \theta_0))\big]) \tag{10}$$

*where $\hat{\mathbb{V}}$ denotes the empirical variance with some fixed sample size.*

The proof can be found in Appendix A.1.1. While only the bound $\sup_t \|\Theta_{\theta_0} - \Theta_{\theta_t}\|_F = \mathcal{O}(\frac{1}{\sqrt{h}})$ has been proven in previous works [25], many empirical studies including those in the present work (see Appendix Fig. 5, Table 3) have shown that the bound decreases faster in practice, on the order of $\mathcal{O}(h^{-1})$ [25, 38]. Note that this result suggests the approximation provided in Proposition 2.2 is *as good as it gets* for describing the predictive variance of non-linearly trained ensembles: the higher order terms would be of a smaller order of magnitude than the non-linear correction to the training, rendering any finer approximation pointless.

## 3 Disentangling deep ensemble variance in practice

The goal of this Section is to validate our theoretical findings in experiments. First, we aim to show qualitatively and quantitatively that the variance of linearly trained neural networks is well approximated by the decomposition introduced in Proposition 2.2. To do so, we investigate ensembles of linearly trained models and analyze their behavior in toy models and on common computer vision classification datasets. We then extend our analyses to *fully-trained* non-linear deep neural networks optimized with (stochastic) gradient descent in parameter space. Here, we confirm empirically the strong influence of the variance description of linearly trained models in these less restrictive settings while being trained to very low training loss. Therefore we showcase the improved understanding of deep ensembles through their linearly trained counterpart and highlight the practical relevance of our study by observing significant OOD detection performance differences of models when removing noise sources in various settings.

## 3.1 Disentangling noise sources in *kernel* models

To isolate the different terms in Proposition 2.2, we construct, from a given initialization $\theta_0$ with the associated linearized model $f^{lin}$, three additional linearly trained models:

$$f^{\text{lin-c}}(x) = \mathcal{Q}_{\theta_0}(x, \mathcal{X})\mathcal{Y}$$
$$f^{\text{lin-a}}(x) = f(x, \theta_0) + \bar{\mathcal{Q}}(x, \mathcal{X})(\mathcal{Y} - f(\mathcal{X}, \theta_0))$$
$$f^{\text{lin-i}}(x) = g(x, \theta_0) + \bar{\mathcal{Q}}(x, \mathcal{X})(\mathcal{Y} - g(\mathcal{X}, \theta_0))$$

where $g(x, \theta_0) = \Theta_{\theta_0}(x, \mathcal{X})\bar{\Theta}(\mathcal{X}, \mathcal{X})^{-1}f(\mathcal{X}, \theta_0)$. Note that the predictive variance over random initialization of these functions corresponds to respectively $\mathbb{V}^c, \mathbb{V}^a, \mathbb{V}^i$ as defined in Section 2.1.

As one can see, we can simply remove the initialization noise from $f^{\text{lin}}$ by subtracting the initial (noisy) function $f(x, \theta_0)$ before training resulting in a *centered* model $f^{\text{lin-c}}$. Equivalently, we can remove noise that originates from the kernel by using the empirical *average* over kernels resulting in model $f^{\text{lin-a}}$. Finally, we can isolate $f^{\text{lin-i}}$ by the same averaging trick as in $f^{\text{lin-a}}$ but use as functional noise $g(x, \theta_0)$ which can be precomputed and added to $f^{\text{lin-c}}$ before training. Note that we neglect the terms involving covariance terms and focus on the parts which are easy to isolate, for linearly trained as well as for standard models. This will later allow us to study practical ways to subtract important parts of the predictive distribution for neural networks leading for example to significant OOD detection performance differences. Now we explore the differences and similarities of these disentangled functions and their respective predictive distributions.

### 3.1.1 Visualizations on a star-shaped toy dataset

To qualitatively visualize the different terms, we construct a two-way star-shaped regression problem on a 2d-plane depicted in Figure 1. After training an ensemble we visualize its predictive variance on the input space. Our first goal is to visualize qualitative differences in the predictive variance of ensembles consisting of $f^{\text{lin}}$ and the 3 disentangled models from above. We train a large ensemble of size 300 where each model is a one-layer ReLU neural network with hidden dimension 512 and 1 hidden layer. As suggested analytically for one hidden layer ReLU networks (see Appendix A.4.3), for example $\mathbb{V}[f^{\text{lin-c}}(x)]$ depends on the angle of the datapoints while $\mathbb{V}[f^{\text{lin}}(x)]$ depicts a superposition of the 3 isolated variances. While the ReLU activation does not satisfy the Lipschitz-continuity assumption of Proposition 2.3, we use it to illustrate and validate our analytical description of the inductive biases induced by the different variance terms. We use the Softplus activation which behaved similarly to ReLU in the experiments in the next Section.

### 3.1.2 Disentangling *linearly trained / kernel* ensembles for MNIST and CIFAR10

Next, we move to a quantitative analysis of the asymptotic behavior of the various variance terms, as we increase the hidden layer size. In Figure 2, we analyze the predictive variance of the kernel models based on MLPs and Convolutional Neural Networks (CNN) for various depths and widths and on subsets of MNIST [39] and CIFAR10. As before, we construct a binary classification task through a MSE loss with dataset size of $N = 100$ and confirm, shown in Figure 2, that $\mathbb{V}^c, \mathbb{V}^i$ decay by $1/h$ over all of our experiments. Crucially, we see that they contribute to the overall variance $\mathbb{V}$ even for relatively large widths. We further observe a decay in $1/h^2$ of the residual term as predicted by Proposition 2.2. As in all of our experiments, the variance magnitude and therefore the influence on $f^{\text{lin}}$ of the disentangled parts is highly architecture and dataset-dependent. Note that the small size of the datasets comes from the necessity to compute the inverse of the kernels for every ensemble member, see Appendix B for a additional analysis on larger datasets and scaling plots of $\mathbb{V}^{cor}$.

In Table 1, we quantify the previously observed qualitative difference of the various predictive variances by evaluating their performance on out-of-distribution detection tasks, where high predictive variance is used as a proxy for detecting out-of-distribution data. We focus our attention on analysing $\mathbb{V}[f^{\text{lin-c}}(x)]$ and $\mathbb{V}[f^{\text{lin-a}}(x)]$, as they are the variance terms containing purely the functional and kernel noise, respectively. As an evaluation metric, we follow numerous studies and compute the area under the receiver operating characteristics curve (AUROC, c.f. Appendix B). We fit a linearized ensemble on a larger subset of the standard 10-way classification MNIST and CIFAR10 datasets using MSE loss. When training our ensembles on MNIST, we test and average the OOD detection performance on FashionMNIST (FM) [40], E-MNIST (EM) [41] and K-MNIST (KM) [42]. When training our ensembles on CIFAR10, we compute the AUROC for SVHN [43], LSUN [44], TinyImageNet (TIN)

Table 1: Test set accuracy and AUROC for deep ensembles of size 10 of MLPs ($h = 1024$, $L = 3$) and CNNs ($h = 256$, $L = 3$) trained on a subset (N=1000) and on of full MNIST and CIFAR10. We indicate small standard deviations $\sigma < 0.005$ obtained over ensemble size (E) with $\pm.00$. In all experiments, the various disentangled models show significant differences in behavior. All linearly trained models follow the gradient descent models behavior tightly. When optimizing with SGD, isolating initial noise sources still affect the ensemble behavior significantly and can lead to improved OOD detection as well as test set accuracy.

| Model | CNN, CIFAR10, N=1000, E=10 | | | | MLP, MNIST, N=1000, E=30 | | | |
|---|---|---|---|---|---|---|---|---|
| | Test (%) | SVHN | LSUN | iSUN | Test (%) | FM | EM | KM |
| $f^{\text{lin}}$ | $36.43^{\pm.90}$ | $.532^{\pm.006}$ | $.809^{\pm.004}$ | $.783^{\pm.004}$ | $91.53^{\pm.40}$ | $.962^{\pm.006}$ | $.922^{\pm.000}$ | $.982^{\pm.001}$ |
| $f^{\text{lin-c}}$ | $37.2^{\pm.44}$ | $.567^{\pm.006}$ | $.693^{\pm.001}$ | $.674^{\pm.004}$ | $89.67^{\pm.15}$ | $.935^{\pm.005}$ | $.881^{\pm.003}$ | $.967^{\pm.002}$ |
| $f^{\text{lin-a}}$ | $30.90^{\pm.53}$ | $.510^{\pm.006}$ | $.764^{\pm.003}$ | $.738^{\pm.000}$ | $91.27^{\pm.06}$ | $.978^{\pm.003}$ | $.922^{\pm.001}$ | $.987^{\pm.000}$ |
| $f^{\text{lin-i}}$ | $32.85^{\pm.21}$ | $.591^{\pm.003}$ | $.683^{\pm.001}$ | $.660^{\pm.000}$ | $91.60^{\pm.42}$ | $.970^{\pm.004}$ | $.908^{\pm.001}$ | $.983^{\pm.002}$ |
| $f^{\text{gd}}$ | $39.70^{\pm.52}$ | $.516^{\pm.002}$ | $.789^{\pm.003}$ | $.763^{\pm.004}$ | $91.43^{\pm.49}$ | $.971^{\pm.005}$ | $.924^{\pm.001}$ | $.986^{\pm.001}$ |
| $f^{\text{gd-c}}$ | $37.47^{\pm.49}$ | $.562^{\pm.004}$ | $.691^{\pm.004}$ | $.670^{\pm.002}$ | $89.67^{\pm.06}$ | $.937^{\pm.005}$ | $.884^{\pm.003}$ | $.968^{\pm.002}$ |
| $f^{\text{gd-a}}$ | $30.53^{\pm1.15}$ | $.509^{\pm.004}$ | $.758^{\pm.005}$ | $.734^{\pm.003}$ | $90.73^{\pm.32}$ | $.978^{\pm.003}$ | $.922^{\pm.001}$ | $.987^{\pm.000}$ |
| $f^{\text{gd-i}}$ | $31.20^{\pm.14}$ | $.583^{\pm.000}$ | $.656^{\pm.003}$ | $.638^{\pm.003}$ | $90.65^{\pm.35}$ | $.977^{\pm.003}$ | $.913^{\pm.002}$ | $.987^{\pm.002}$ |

| Model | CNN, CIFAR10, N=50000, E=5 | | | | MLP, MNIST, N=50000, E=5 | | | |
|---|---|---|---|---|---|---|---|---|
| | Test (%) | SVHN | LSUN | iSUN | Test (%) | FM | EM | KM |
| $f^{\text{sgd}}$ | $62.68^{\pm.36}$ | $.557^{\pm.01}$ | $.884^{\pm.00}$ | $.864^{\pm.00}$ | $95.70^{\pm.12}$ | $.974^{\pm.005}$ | $.930^{\pm.001}$ | $.991^{\pm.001}$ |
| $f^{\text{sgd-c}}$ | $57.03^{\pm.14}$ | $.554^{\pm.00}$ | $.791^{\pm.00}$ | $.781^{\pm.00}$ | $94.43^{\pm.01}$ | $.924^{\pm.016}$ | $.873^{\pm.006}$ | $.962^{\pm.004}$ |
| $f^{\text{sgd-a}}$ | $58.83^{\pm.22}$ | $.455^{\pm.00}$ | $.864^{\pm.00}$ | $.845^{\pm.00}$ | $97.48^{\pm.13}$ | $.988^{\pm.002}$ | $.943^{\pm.001}$ | $.995^{\pm.001}$ |

and CIFAR100 (C100), see Appendix Table 4 for the variance magnitude and AUROC values for all datasets.

The results show significant differences in variance magnitude and AUROC values. While we do not claim competitive OOD performance, we aim to highlight the differences in behavior of the isolated functions developed above: we see for instance that for (MLP, MNIST, N=1000), $f^{\text{lin-a}}$ generally performs better than $f^{\text{lin}}$ in OOD detection. Indeed, the overall worse performance of $\mathbb{V}[f^{\text{lin-c}}(x)]$ seems to be affecting that of $\mathbb{V}[f^{\text{lin}}(x)]$ which contains both terms. On the other hand, we see that for the setup (CNN, CIFAR10, N=1000) $\mathbb{V}[f^{\text{lin}}(x)]$ is not well described by this interpolation argument, which highlights the influence of the other variance terms described in Proposition 2.2. Furthermore, the OOD detection capabilities of each function seem to be highly dependent on the particular data considered: Ensembles of $f^{\text{lin-c}}$ are relatively good at identifying SVHN data as OOD, while being poor at identifying LSUN and iSUN data. These observations highlight the particular inductive bias of each variance term for OOD detection on different datasets.

We further report the test set generalization of the ensemble mean of different functions, highlighting the diversity in the predictive mean of these models as well. Note that for $N >= 1000$ we trained the ensembles in linear fashion with gradient flow (which coincides with the kernel expression) up until the MSE training error was smaller than $0.01$.

### 3.2 Does the refined variance description generalize to standard gradient descent in practice?

In this Section, we start with empirical verification of Proposition 2.3 and show that the bound in equation 10 holds in practice. Given this verification, we then propose equivalent disentangled models as those previously defined but in the non-linear setting, and 1) show significant differences in their predictive distribution but also 2) investigate to which extent improvements in OOD detection translate from kernel / linearly to fully non-linearly trained models. We stress that we do not consider early stopped models and aim to connect the kernel with the gradient descent models faithfully.

#### 3.2.1 Survival of the kernel noise after training

To validate Proposition 2.3, we first introduce $f^{\text{gd}}(x) = f(x, \theta_t)$, a model trained with standard gradient descent of $t$ steps i.e. $\theta_t = \theta_0 - \sum_{i=0}^{t-1} \eta \nabla_\theta f(\mathcal{X}, \theta_i)(\mathcal{Y} - f(\mathcal{X}, \theta_i))$. To empirically verify

Proposition 2.3, we introduce the following ratio

$$\mathcal{R}(f) = \exp\left(\mathbb{E}_{x\sim\mathcal{X}'}\left(\log[\frac{\|\hat{\mathbb{V}}[f^{\text{lin}}(x)] - \hat{\mathbb{V}}[f^{\text{gd}}(x)]\|}{\|\hat{\mathbb{V}}^c(x) + \hat{\mathbb{V}}^i(x)\|}]\right)\right) \tag{11}$$

where the empirical variances are computed over random initialization, and the expectation over some data distribution which we choose to be the union of the test-set and the various OOD datasets. Given a datapoint $x$, the term inside the $\log$ measures the ratio between the discrepancy of the variance between the linearized and non-linear ensemble, against the refined variance terms. $\mathcal{R}(f)$ is then the geometric mean of this ratio over the whole dataset. Proposition 2.3 predicts that the ratio remains bounded as the width increases, suggesting that the refined terms contribute to the final predictive variance of the non-linear model in a non negligible manner. We empirically verify this prediction for various depths in Fig. 3 and Appendix Figure 6, for functions trained on subsets MNIST and CIFAR10. Note that for all our experiments we also empirically verify the assumption from Proposition 2.3 (see Appendix Figure 5, Table 3).

### 3.2.2 Disentangling noise sources in *gradient descent* non-linear models

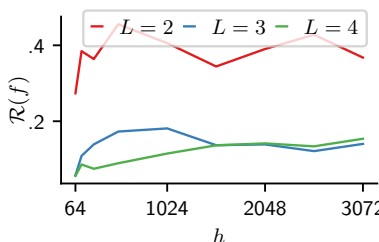

Motivated by the empirical verification of Proposition 2.3, we now aim to isolate different noise sources in non-linear models trained with gradient descent. Starting from a non-linear network $f^{\text{gd}}$, we follow the same strategy as before and silence the functional initialization noise by *centering* the network (referred as $f^{\text{gd-c}}(x)$) by simply subtracting the function at initialization. On the other hand, we remove the kernel noise with a simple trick: We first sample a random weight $\theta_0^c$ once, and use it as the weight initialization for all ensemble members. A function noise is added by first removing the function initialization from $\theta_0^c$, and adding that of a second random network which is not trained. The

Figure 3: $\mathcal{R}(f)$ of MLPs with multiple widths and depths ($L \in \{2, 3, 4\}$) trained on a subset (100) of MNIST . As predicted, we observe $\mathcal{R}(f)$ bounded as we increase the width in support of our theoretical analysis.

resulting functions (referred as $f^{\text{gd-a}}(x)$) will induce and ensemble which will only differ in their functional initialization while having the same Jacobian

$$f^{\text{gd-c}}(x) = f(x, \theta_t) - f(x, \theta_0),$$
$$f^{\text{gd-a}}(x) = f(x, \theta_t^c) - f(x, \theta_0^c) + f(x, \theta_0).$$

We furthermore introduce $f^{\text{gd-i}}(x)$, the non linear counterpart to $f^{\text{lin-i}}(x)$, which we construct similarly to $f^{\text{gd-a}}(x)$ but using $g(x, \theta_0, \theta_0^c) = \Theta_{\theta_0}(x, \mathcal{X})\Theta_{\theta_c}(\mathcal{X}, \mathcal{X})^{-1}f(\mathcal{X}, \theta_0)$ as the function initialization instead of $f(x, \theta_0)$ (see Section 2.1 and Appendix A.3.1 for the justification). Unlike $f^{\text{gd-a}}$ and $f^{\text{gd-c}}$, constructing $f^{\text{gd-i}}$ requires the inversion of large matrices due to the way $g$ is defined, a challenging task for realistic settings. While its practical use is thus limited, we introduce it to illustrate the correspondence of correspondence of the predictive variance of linearized vs non-linear deep ensemble.

Given these simple modifications of $f^{\text{gd}}$, we rerun the experiments conducted for the linearly trained models for moderate dataset sizes (N=1000). We observe close similarities in the OOD detection capabilities as well as predictive variance between the introduced non-linearly trained ensembles and their linearly trained counterparts. We further train these models on the full MNIST dataset (N=50000) for which we show the same trend in Appendix Table 5. We also include the ensemble' performance when trained on the full CIFAR10 dataset. Intriguingly, the relative performance of the ensemble is somewhat preserved in both settings between N=1000 and N=50000, even when training with SGD, promoting the use of quick, linear training on subset of data as a proxy for the OOD performance of a fully trained deep ensemble.

Similar to the case of (MLP, MNIST, N=1000/50000), we observe that $f^{\text{gd}}$ ensemble performance is an interpolation of $f^{\text{gd-c}}$ and $f^{\text{gd-a}}$ which interestingly performs often favorably, on different OOD data. To understand if the noise introduced by SGD impacts the predictive distribution of our disentangled ensembles, we compared the behavior of $f^{\text{gd}}$ and $f^{\text{sgd}}$ in the lower data regime

of $N = 1000$. Intriguingly, we show in Appendix Table 6 that no significant empirical difference between GD and SGD models can be observed and hypothesize that noise sources discussed in this study are more important in our approximately linear training regimes. To speed up experiments we used (S)GD with momentum (0.9) in all experiments of this subsection.

### 3.2.3 Removing noise of models possibly far away from the linear regime

Finally, we investigate the OOD performance of the previously introduced model variants $f^{\text{sgd}}$, $f^{\text{sgd-c}}$ and $f^{\text{sgd-d}}$ in more realistic settings. To do so we train the commonly used WideResNet 28-10 [45] on CIFAR10 with BatchNorm [46] Layers and cross-entropy (CE) loss with batchsize of 128, without data augmentation (see Table 3.2.3). These network and training algorithm choices are considered crucial to achieving state-of-the-art and superior performance compared to their linearly trained counterparts. Strikingly, we notice that our model variants, which each isolate a different *initial* noise source, significantly affect the OOD capabilities of the *final* models when the training loss is virtually 0 - as in all of our experiments. This indicates that the discussed noise sources influence the ensemble's predictive variance long throughout training. We provide similar results for CIFAR100 and FashionMNIST in Table B.1 and B.1 of the Appendix B. We stress that we do not claim that our theoretical assumptions hold in this setup.

Table 2: Test set accuracy and AUROC for WRN 28-10 ensembles of size 8 trained on CIFAR10 on the cross entropy (CE) or MSE loss. While the models perform similarly on test set, a significant advantage of $f^{\text{sgd-c}}$ in OOD detection is observed across most OOD datasets. Standard deviations $\sigma$ computed over 5 seeds are indicated with $\pm$. In bold are values that outperform $f^{\text{sgd}}$ with $p < 0.2$.

| Model | Loss | Test (%) | C100 | SVHN | LSUN | TIN | iSUN |
|-------|------|----------|------|------|------|-----|------|
| $f^{\text{sgd}}$ | CE | $89.36^{\pm 0.36}$ | $0.830^{\pm 0.001}$ | $0.900^{\pm 0.002}$ | $0.891^{\pm 0.002}$ | $0.860^{\pm 0.001}$ | $0.883^{\pm 0.001}$ |
| $f^{\text{sgd-c}}$ | CE | $89.56^{\pm 0.30}$ | $\mathbf{0.831}^{\pm 0.003}$ | $0.899^{\pm 0.004}$ | $\mathbf{0.895}^{\pm 0.003}$ | $\mathbf{0.862}^{\pm 0.003}$ | $\mathbf{0.885}^{\pm 0.003}$ |
| $f^{\text{sgd-a}}$ | CE | $89.01^{\pm 0.32}$ | $0.827^{\pm 0.002}$ | $0.894^{\pm 0.003}$ | $0.887^{\pm 0.003}$ | $0.855^{\pm 0.004}$ | $0.879^{\pm 0.002}$ |
| $f^{\text{sgd}}$ | MSE | $77.94^{\pm 0.22}$ | $0.739^{\pm 0.004}$ | $0.863^{\pm 0.006}$ | $0.823^{\pm 0.007}$ | $0.795^{\pm 0.007}$ | $0.813^{\pm 0.008}$ |
| $f^{\text{sgd-c}}$ | MSE | $77.88^{\pm 0.30}$ | $0.739^{\pm 0.001}$ | $\mathbf{0.880}^{\pm 0.006}$ | $\mathbf{0.829}^{\pm 0.005}$ | $\mathbf{0.807}^{\pm 0.006}$ | $0.813^{\pm 0.005}$ |
| $f^{\text{sgd-a}}$ | MSE | $75.34^{\pm 0.18}$ | $0.707^{\pm 0.004}$ | $0.841^{\pm 0.011}$ | $0.784^{\pm 0.003}$ | $0.763^{\pm 0.010}$ | $0.761^{\pm 0.002}$ |

## 4 Conclusion

The generalization on in-and out-of-distribution data of deep neural network ensembles is poorly understood. This is particularly worrying since deep ensembles are widely used in practice when trying to asses if data is out-of-distribution. In this study, we try to provide insights into the sources of noise stemming from initialization that influence the predictive distribution of trained deep ensembles. By focusing on large-width models we are able to characterize two distinct sources of noise and describe an analytical approximation of the predictive variance in some restricted settings. We then show theoretically and empirically how parts of this refined predictive variance description in the linear training regime survive and impact the predictive distribution of non-linearly trained deep ensembles. This allows us to extrapolate insights of the tractable linearly trained deep ensembles into the non-linear regime which can lead to improved out-of-distribution detection of deep ensembles by eliminating potentially unfavorable noise sources. Although our theoretical analysis relies on the closeness to linear gradient descent which has shown to result in less powerful models in practice, we hope that our surprising empirical success of noise disentanglement sparks further research into using the lens of linear gradient descent to understand the mysteries of deep learning.

## Acknowledgments and Disclosure of Funding

Seijin Kobayashi was supported by the Swiss National Science Foundation (SNF) grant CR-SII5_173721. Pau Vilimelis Aceituno was supported by the ETH Postdoctoral Fellowship program (007113). Johannes von Oswald was funded by the Swiss Data Science Center (J.v.O. P18-03). We thank Christian Henning, Frederik Benzing and Yassir Akram for helpful discussions. Seijin Kobayashi and Johannes von Oswald are grateful for Angelika Steger's and João Sacramento's overall support and guidance.

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
