# OpenReview forum: "Disentangling the Predictive Variance of Deep Ensembles through the Neural Tangent Kernel"
_NeurIPS.cc/2022/Conference — NeurIPS 2022 Accept_

### Official Review · Reviewer_GBTo · 2022-07-09

**Rating:** 7
**Confidence:** 3
**Soundness:** 4 excellent
**Presentation:** 3 good
**Contribution:** 3 good

**Summary:**

This manuscript uses the concepts of Neural Network Gaussian Processes and Neural Tangent Kernels to analyze the sources of prediction variance from neural networks.  By making several reasonable assumptions, they can break the predictive variance into the superposition of several distinct terms.  Through this mathematical analysis and accompanying empirical studies, they demonstrate the practical consequences of this decomposition on several different neural networks.  Finally, they demonstrate that these decompositions can be used to build a greater intuitive understanding on out-of-distribution data.

**Questions:**

How exactly were OOD tasks performed?  Update: This has been largely clarified after the rebuttal.

I do not fully understand the claims about the relationship between each variance term and OOD on data-specific performance.  Can you please elaborate on your claims of how the results highlight and explain these relationships?  Update: The response really highlights the challenges of interpretation.  Further claims and exploration are beyond the scope of this work, but I would encourage the authors to continue examining this question.

**Limitations:**

I believe that the authors have been forthright and fair in their descriptions of the limitations of their theoretical analysis.

**Strengths And Weaknesses:**

This manuscript clearly presents its theoretical arguments and accompanies them with empirical studies that confirm the findings.  Given the large focus on robustness to out-of-distribution data in recent years, this type of decomposition can help practitioners think about potential ways to understand and improve their models.  I believe that this decomposition is original, and has high quality and significance.

The final empirical studies, though, are underexplained and a bit confusing.  It is not clearly described enough what prediction tasks are being performed (simply “out-of-distribution” tasks) and what approach is being used to determine out-of-distribution samples.  This needs to be much more clearly described in a revised manuscript.  The claims about performance are not refined enough. They claim that “These observations highlight the particular inductive bias of each variance term for OOD detection.” However, it is not clearly described enough how these inductive biases relate to the particular predictive performance.  The analysis would be much enhanced with clearer statements about exactly how these inductive biases relate to performance.

---

> ### Author Response · Authors · 2022-08-02
> **Response to Reviewer GBTo**
>
> We thank the reviewer for the effort of evaluating our work carefully and the overall positive feedback. We share the enthusiasm of the reviewer that the provided work can provide useful insights into the inner workings of deep ensembles, especially its OOD detection mechanisms. We hope that we can address the remaining concerns below.
>
> **P1**:  The final empirical studies, though, are underexplained and a bit confusing. It is not clearly described enough what prediction tasks are being performed (simply “out-of-distribution” tasks) and what approach is being used to determine out-of-distribution samples. This needs to be much more clearly described in a revised manuscript.
>
> **A1**:  In order to clarify how we compute the AUROC values in detail, we will add an additional appendix section that provides a detailed description of the procedure including pseudo-code. As these metrics are essential to the paper, we thank and agree with the reviewer that this needs a precise description.
>
> **P2**: The claims about performance are not refined enough. They claim that “These observations highlight the particular inductive bias of each variance term for OOD detection.” However, it is not clearly described enough how these inductive biases relate to the particular predictive performance. The analysis would be much enhanced with clearer statements about exactly how these inductive biases relate to performance.
>
> **A2**:  As also mentioned in the general response, we will make this point much clearer in the final manuscript. First, it should be noted that we provide a closed-form analytic description of the predictive variances for one layer hidden ReLU neural networks (see also Figure 7) which indicates that in some limit, the predictive variance is dictated by the angle between the data for $V^c$. Although we do not provide similar results for deeper and/or convolutional neural network models, we argue that similar (but more complicated) descriptions lead to significantly different models resulting in vast differences in variances and therefore OOD detection performance. We illustrate this point by drawing a correspondence between the OOD detection performance of a simple angle-based and L2-distance based baselines and that of $V^c$ and $V^a$ in the general response.
>
> Note that it is difficult to describe analytically or even informally how the different design choices in deep learning generally affect model behavior, and therefore also the predictive variances of deep ensembles. Nevertheless, we hope to provide at least some practically relevant and meaningful insights through our work, and believe our refined large-scale results from the general response to be convincing.
>
> **Q2**:  How exactly were OOD tasks performed. I do not fully understand the claims about the relationship between each variance term and OOD on data-specific performance. Can you please elaborate on your claims of how the results highlight and explain these relationships?
>
> **A3**: If this point is not answered sufficiently in **A2** and by the additional angle vs. distance OOD detection analyses in the general response, please let us know. We are open for any further discussion.
>
> We again thank the reviewer for the constructive feedback, and hope we adequately addressed their questions. We are happy to discuss any additional points during the rebuttal period.

---

### Official Review · Reviewer_e4Rz · 2022-07-12

**Rating:** 7
**Confidence:** 3
**Soundness:** 2 fair
**Presentation:** 2 fair
**Contribution:** 3 good

**Summary:**

This paper proposes to understand finite-width neural networks in terms of linearly trained finite width networks. The authors propose to decompose the variance of ensembles of linearly trained, finite width neural networks into different sources (of noise), and study empirically the implications of this decomposition for understanding actual finite-width networks trained with gradient descent and stochastic gradient descent. The work builds from [1][2] which proposes to approximate a sufficiently wide network $f_\theta$ with its linearly trained counterpart $f_lin$, and in the infinite with limit with the fully trained network $f_\inf$.
The paper decompose the variance of $f_\inf$  and $f_lin$, and interprets different variance terms arising in ensembles of linear predictors in terms of kernel and functional noise (ln 140), which affects the predictive variance. The derivation relies on certain independence assumptions that do not hold in practice.
The paper additionally claims ways to eliminate sources of noise which improve OOD detection in ensembles. The results comparing the networks which only include disentangled noise sources show no consensus in how/if the proposed sources of noise affect the total variance. The claims of improving OOD detection after reducing sources of noise coming from the results don’t take into account that the results lie within the confidence interval (cf. Table 2).

**Questions:**

- what is it the implication of assuming the functional initialization noise is decorrelated from the $f(x)$ in proposition 2.1?
- In line 140, why is it obvious that Va corresponds to the inf width model?
- What are the implications of Fig 1, if we cannot draw any conclusions from the shape/location of the data points, as the authors mentioned this changes per model/dataset?
- Could we add a Fig 2 which includes the baseline, total V? this would be useful to better understand the scale of different V.


**Strengths And Weaknesses:**

Strengths:
This paper proposes to understand finite-width neural networks in terms of linearly trained finite width networks. The paper decompose the variance of ensembles of linearly trained, finite width neural networks into different sources (of noise). The work builds from  [1][2] which proposes to approximate a sufficiently wide network $f_\theta$ with its linearly trained counterpart $f_lin$, and in the infinite with limit with the fully trained network $f_\inf$. This is an interesting perspective for analyzing deep ensemblesl

Weaknesses:
- The paper decompose the variance of ensembles of $f_\inf$  and $f_lin$.  The paper claims the variance for the ensemble of $f_lin$  includes two components which correspond to the functional initialization noise and the other to the kernel noise (ln 140), and that this noise affects the predictive variance. During the derivation it is assumed that the that the functional initialization noise is decorrelated from the $f(x)$ (ln 133). The authors mentioned that this is future work but this is a central assumption in this work.
- The results comparing the networks which only include disentangled noise sources show no consensus in how/if the proposed sources of noise affect the total variance. Could we add a Fig 2 which includes the baseline, total V?
- The claim of improving OOD detection after reducing sources of noise coming from the results in Table 2. In Table 2, we can see that the comparisons lie within each other's confidence intervals, so it is unclear why we can claim a model is better than another.
- Some sections need additional clarification, i.e. the description of the variance components in paragraph ln 147 to 160 is not clear to me. Given that this paper is offering a new perspective on the variance decomposition a clear description of each component would be greatly beneficial. I had to come back to this twice. The same holds for the implications of the results in Fig 1.
- The assumptions and implications of the propositions, in particular 2.1 and 2.3, could be clearer.  Proposition 2.3 can be true for a single network, is this correct? This proposition is in the deep ensemble section 2.2.

---

> ### Author Response · Authors · 2022-08-02
> **Response to Reviewer e4Rz**
>
> We thank the reviewer for carefully evaluating our work and providing constructive comments on the paper. We will address the concerns one by one and hope that the additional clarification and the provided analyses will lead the reviewer to vote for an acceptance of the paper. We are happy to provide any additional information upon the request of the reviewer.
>
> **P1**: The paper decomposes the variance of ensembles of f_inf and f_lin. The paper claims the variance for the ensemble of f_lin includes two components which correspond to the functional initialization noise and the other to the kernel noise (ln 140), and that this noise affects the predictive variance. During the derivation it is assumed that the that the functional initialization noise is decorrelated from the f(x) (ln 133). The authors mentioned that this is future work but this is a central assumption in this work.
>
> **A1**:  This is a great point, and we agree with the reviewer that this crucial assumption needs an improved detailed discussion about its consequences and limitations.
>
> In short, if the assumption is not satisfied, $\mathbb{V}^{\text{corr}}$ from Proposition 2.2 becomes much more complex, involving 4th moments of the finite width NNGP kernel fluctuation. This is challenging to analytically compute even for a single hidden layer neural network, which lead us to make the simplifying assumption. Nevertheless, the presence of the other variance terms in the total variance, their analytical expression and scaling behavior still hold.
>
> More importantly, together with Proposition 2.3, the use of linearized deep ensemble to approximate the predictive variance of a deep ensemble remains justified, and in the particular case of deep ensemble of centered models ($f^{\text{gd-c}}$), our description $V^c$ is in fact the first order approximation of its predictive variance, irrespective of the aforementioned assumption. We think our empirical results in Table1 support this claim, and hope that the new experimental results are convincing to the reviewer in further demonstrating the effect these initial noise terms have in fully trained nonlinear deep ensemble predictive variance, as well as the meaningful application which can be deduced.
>
> We will add all of these important points in the discussion of the proposition as well as add these limitations in the conclusion of the final manuscript.

---

> > ### Author Response · Authors · 2022-08-02
> > **Response to Reviewer e4Rz  pt 2**
> >
> > **P2**: The results comparing the networks which only include disentangled noise sources show no consensus in how/if the proposed sources of noise affect the total variance. Could we add a Fig 2 which includes the baseline, total V?
> >
> > **A2**: Thank you for pointing out this important baseline. We agree with the reviewer that these numbers are important and missing and provide the numbers here in Table form. We will adjust the Figure in the final manuscript accordingly.
> >
> > | Width-Depth | $\mathbb{V}[f]$ | $\mathbb{V}[f^{\text{lin-a}}]$ | $\mathbb{V}[f^{\text{lin-i}}]$ | $\mathbb{V}[f^{\text{lin-c}}]$ | $\mathbb{V}[f^{\text{lin-cor}}]$ |
> > |-------------|-----------------|--------------------------------|--------------------------------|--------------------------------|----------------------------------|
> > | 64-2        |           4.063 |                          2.497 |                          0.914 |                          1.080 | 0.161                            |
> > | 128-2       | 3.718           |                          2.646 |                          0.468 |                          0.717 | 0.069                            |
> > | 256-2       | 3.108           |                          2.540 |                          0.239 |                          0.386 | 0.050                            |
> > | 512-2       | 2.730           |                          2.413 |                          0.126 |                          0.217 | 0.016                            |
> > | 1024-2      | 2.640           |                          2.478 |                          0.056 |                          0.113 | 0.020                            |
> > |             |                 |                                |                                |                                |                                  |
> > | 64-3        | 4.094           | 2.300                          | 1.612                          | 1.181                          | 0.524                            |
> > | 128-3       | 3.482           | 2.322                          | 0.670                          | 0.787                          | 0.172                            |
> > | 256-3       | 3.106           | 2.365                          | 0.353                          | 0.481                          | 0.034                            |
> > | 512-3       | 2.823           | 2.389                          | 0.184                          | 0.286                          | 0.019                            |
> > | 1024-3      | 2.622      | 2.382                          | 0.088                          | 0.159                          | 0.015                            |
> >
> > As can be seen, each variance term contributes to the total variance.
> >
> > **P3**:  The claim of improving OOD detection after reducing sources of noise coming from the results in Table 2. In Table 2, we can see that the comparisons lie within each other's confidence intervals, so it is unclear why we can claim a model is better than another.
> >
> > **A3**: This is another great point, and we thank the reviewer for bringing it up. Motivated by this comment, we provide in our general response refined results on the experimental setup, which indeed show some statistically significant improvement for OOD detection on FashionMNIST, CIFAR10 and CIFAR100 - especially for models trained with the MSE loss. We also perform the same study using AlexNet and observe similar trends. The results show that, for most of these applications, $\mathbb{V}^{\text{c}}$ surprisingly seem to have the right inductive bias to perform well.
> >
> > We stress that we do not claim any of our models lead to better OOD detection in general, since this would require a proper definition of OOD data, which is lacking in the literature to the best of our knowledge.
> >
> > **P4**:  Some sections need additional clarification, i.e. the description of the variance components in paragraphs ln 147 to 160 is not clear to me. Given that this paper is offering a new perspective on variance decomposition a clear description of each component would be greatly beneficial. I had to come back to this twice. The same holds for the implications of the results in Fig 1.
> >
> > **A4**: Thank you for raising this issue. We are happy for any suggestions to make our paper more comprehensible and we will work on improving this section in the final manuscript. In particular, we will clearly derive how  $\mathbb{V}^{\text{a}}$ relate to the infinite width variance, as well as clarify the description of  $\mathbb{V}^{\text{i}}$ by moving elements of Appendix A.3.1 to the main text.

---

> > > ### Author Response · Authors · 2022-08-02
> > > **Response to Reviewer e4Rz pt 3**
> > >
> > > **Q1**:  The assumptions and implications of the propositions, in particular 2.1 and 2.3, could be clearer. Proposition 2.3 can be true for a single network, is this correct? This proposition is in the deep ensemble section 2.2.
> > >
> > > **A5**: We agree that this proposition is general. We would like to keep it in the (non linear) deep ensemble section, as it relates to the effect of nonlinear training - but we will clarify the presentation by clearly stating the result holds for single networks.
> > >
> > > **Q2**: What is it the implication of assuming the functional initialization noise is decorrelated from the f(x) in proposition 2.1?
> > >
> > > **A6**: See above **A1**.
> > >
> > > **Q3**: In line 140, why is it obvious that Va corresponds to the inf width model?
> > >
> > > **A7**: One can see that the expression of $\mathbb{V}^{\text{a}}$ is identical to that of $\mathbb{V}^{\text{inf}}$ except for using the expected NNGP kernel (for finite width) and NTK kernel instead of the infinite width counterparts. In the case of a one hidden layer network, the two variances become identical since the expected finite width NNGP kernel or NTK kernel is equal to their infinite width counterparts based on the central limit theorem. We will add a section in the appendix explaining this correspondence in detail.
> > >
> > > **Q4**: What are the implications of Fig 1, if we cannot draw any conclusions from the shape/location of the data points, as the authors mentioned these changes per model/dataset?
> > >
> > > **A8**: Our motivation for showing Figure 1 is the following.
> > >
> > > 1. We want to provide an intuition of how drastic the different inductive biases induced by our disentangled models can be. As described above, we will work carefully on improving our manuscript to make the motivations and take-always of our empirical section and paper in general clearer.
> > >
> > > 2. The figure validates the closed-form description of one hidden layer neural network predictive variance, especially the angle and distance description derived in the Appendix.
> > >
> > > We thank the reviewer again for the feedback which helped us to strongly strengthens the paper. We believe our additional empirical results to be convincing, and hope the reviewer agrees to raise the score.

---

> > > > ### Comment · Reviewer_e4Rz · 2022-08-09
> > > > **reviewer response**
> > > >
> > > > Thanks to the authors for their response, they have addressed several of my concerns and questions and I have updated my score accordingly.

---

> > > > > ### Author Response · Authors · 2022-08-09
> > > > > **Thank you again!**
> > > > >
> > > > > Dear reviewer e4Rz,
> > > > > thank you for updating your score and for the encouraging feedback.
> > > > > We believe that the proposed changes, based on your criticism and insightful comments, will lead to a much-improved final manuscript. Thanks again

---

> ### Author Response · Authors · 2022-08-09
> **Happy to provide final clarifications**
>
> Dear reviewer e4Rz,
>
> Many thanks again for your helpful comments and the effort in evaluating our work carefully. We want to politely remind you that the author-reviewer discussion period is shortly coming to an end. Please do not hesitate to contact us before the interactive discussion period ends to discuss any further concerns.

---

### Official Review · Reviewer_CZjD · 2022-07-13

**Rating:** 7
**Confidence:** 2
**Soundness:** 3 good
**Presentation:** 3 good
**Contribution:** 4 excellent

**Summary:**

This paper studies ensembles of linearly trained deep networks, for which a decomposition of the variance is proposed into two main components:
 - V_a results from the variance of the initial value f_theta_0
 - V_c results from the variance of the initial kernel Theta_0
and other components that decay faster as the width of the network is grown.

It is also shown that this decomposition still hold when training non-linearly one hidden layer networks under stability assumption of the NTK during training. It is conjectured (and empirically somehow observed) that the difference in prediction between linear and non-linear training is small enough for the above decomposition to still be relevant in deeper networks.

Experiments on toy datasets enable visualizing the different sources of variance. Experiments on more difficult (MNIST, CIFAR10) datasets show that conclusions vary across setups. A method is proposed to isolate sources of variance in actual setups.

**Questions:**

 - it is not clear that the y scale in figure 2 and 5 is a log scale from just looking at the figure (and not mentioned in the text).
 - in figure 2, contrary to what is mentioned in the text, it looks that the decay of V_c is slightly slower than 1/h (thus suggesting that V_c does not decay in O(1/h)). It is difficult to see because of the log-log scale but it looks to be closer to 1/h^{3/4}
 - in figure 5 in appendix, I did not understand how many steps of GD were run for this experiment. I am guessing that it uses very few steps since the dataset if very small (N=100), and I am expecting the results to be different on actual setups. What do you think?

**Limitations:**

I did not find a discussion about potential negative societal impact, nor do I need that such paper should have one.

The limitations of the setups (linearly trained networks, small datasets) are discussed.

**Strengths And Weaknesses:**

*Originality*: To the best of my knowledge (though I am not so familiar with recent literature), the idea of studying ensemble of ensembles of linearly trained networks is new and sound. The results linking infinite-width models to linearly-trained finite networks are new, in the line of works around the NTK formalism.

*Quality*: The paper is well organised, the experiments and figures are relevant to the discussion. I had some comments about the experiments (see questions below)

*Clarity*: The material is well presented. The theoretical results are clearly discussed, as well as the assumptions. I however was not sure what is the main message to get from the experiments on large networks/datasets. This is probably related to the latter, but I did not understand what are the "practical ways to eliminate noisy sources" claimed to be proposed in the abstract.

*Significance:*: The study of ensembles of neural nets is of particular relevance as many recent papers use ensembles of deep nets. The decomposition between variance of the initial function and variance of the initial kernel is of interest for understanding the training dynamics of neural networks.

---

> ### Author Response · Authors · 2022-08-02
> **Response to Reviewer CZjD**
>
> We thank the reviewer for the effort of evaluating our work carefully and the overall positive feedback. We particularly appreciated the reviewer’s attention to crucial details. We hope the additional results in the general response further strengthen the paper, and that we appropriately address any remaining concerns below.
>
> **P1**:  I however was not sure what is the main message to get from the experiments on large networks/datasets. This is probably related to the latter, but I did not understand what are the "practical ways to eliminate noisy sources" claimed to be proposed in the abstract.
>
> **A1**:  Thank you for bringing up this important point. As also discussed in the general response, we will work on generally clarifying the motivation behind our empirical section and the conclusions one can draw from them in the manuscript. In Table 1, we provide evidence that the tricks we develop to eliminate noise sources in the linear training regime translate to the gradient descent training regime. While drawing the correspondence of the linearized and nonlinear deep ensembles is computationally infeasible when training on large models and dataset as the training time of linearized deep ensembles increases dramatically, our main motivation for moving to larger models was to investigate if these different inductive model biases lead to significantly different predictive variances and therefore OOD detection, even in this realistic deep learning setup.
>
> **Q1**: it is not clear that the y scale in figure 2 and 5 is a log scale from just looking at the figure (and not mentioned in the text).
>
> **A2**: Thank you for spotting this mistake. We will clearly mention that indeed the figure is in log log scale in the final manuscript.
>
> **Q2**: in figure 2, contrary to what is mentioned in the text, it looks that the decay of V_c is slightly slower than 1/h (thus suggesting that V_c does not decay in O(1/h)). It is difficult to see because of the log-log scale but it looks to be closer to 1/h^{3/4}
>
>  **A3**: Thank you for studying our work in detail and making this interesting observation!  Our scaling is asymptotic and we argue that the apparent scaling in ~1/h^(3/4) is due to the finite width effect stemming from the residual of the Taylor approximation, which disappears as we approach bigger width. Motivated by the reviewer’s comment, we conducted a linear regression on the log-log scaled curve which indeed does yield a slope of -0.82 for a two-layer MLP.  Nevertheless, the slope approaches -1 as we discard smaller widths ( -0.88, -0.89,-0.94 for resp. the last 4,3, and 2 widths). We observe similar trends for other depths and architecture and are confident to claim that our decay indeed is asymptotically in O(1/h).
>
> **Q3**:  in figure 5 in appendix, I did not understand how many steps of GD were run for this experiment. I am guessing that it uses very few steps since the dataset if very small (N=100), and I am expecting the results to be different on actual setups. What do you think?
>
>  **A4**: Thank you again for thoroughly studying our work. We train the models until the largest training loss over the training samples is less than 0.01, which still results in usually thousands of update steps even for a hundred datapoints. Due to the computational complexity involved with computing the NTK, we cannot extend this analysis to very large datasets - but we are actively extending it to N=1000. We will update you with our results during the discussion period.
>
> We again thank the reviewer for the constructive feedback, and hope we adequately addressed the reviewer’s questions. We are happy to discuss any additional points during the rebuttal period.

---

> > ### Author Response · Authors · 2022-08-07
> > **Follow up result**
> >
> > We provide the requested additional result below. Computing the NTK for a 10-way classification with N=1000 data is computationally intensive, and due to time and resource constraints, we only provide the result for N=500 here. We will however include the full result in the final script.
> >
> > | L | h   | $\|\|\theta_0-\theta_t\|\|/\|\|\theta_0\|\|$ |
> > |---|------|--------------------------------------------------------|
> > | 2 | 512 |                                             0.2213703 |
> > | 2 | 1024 |                                             0.1406264 |
> > | 2 | 2048 | 0.0675842                                             |
> > | 2 | 4096 | 0.0367682                                             |
> > |  |     |                                                       |
> > | 3 | 512 | 0.35                                                  |
> > | 3 | 1024 | 0.22183615                                            |
> > | 3 | 2048 | 0.12262867                                            |
> > | 3 | 4096 | 0.07011652                                            |
> >
> > As can be seen, the trend is roughly preserved even for a larger data set. For L=2, the scaling is clearly 1/h, while for L=3, it is slower, yet clearly quicker than 1/sqrt(h), which is sufficient for our result to hold.
> >
> > We thank the reviewer again for the feedback, and are more than happy to address any additional questions or requests.

---

### Official Review · Reviewer_rqPL · 2022-07-16

**Rating:** 7
**Confidence:** 2
**Soundness:** 3 good
**Presentation:** 2 fair
**Contribution:** 4 excellent

**Summary:**

This work is focused on decomposing the sources of variance in neural network predictions - the predictive variance of neural networks is an important quantity that allows for neural ensembles (and other approximate bayesian models) to detect OOD samples without seeing OOD data. However, this quantity is poorly understood - this work proposes to take a close, principled look at neural network predictive variance by studying _linear_ neural networks. Linear neural network predictions can be represented via a kernel expression, allowing a theoretical decoupling of model predictions and (noisy) gradient descent.

The work largely comprises of 2 main sections. The first section (section 2) contains the main theoretical results of the work, and begins by introducing linear neural networks and infinitely wide neural networks. The work then builds up a theoretical decomposition of variance terms in the predictive distribution of an ensemble of neural networks. To do this, the authors begin by analyzing an ensemble of infinitely wide neural networks, whose kernels are deterministic, demonstrating that it's predictive variance can explained by the _functional noise_ over different initializations. The authors then extend this analysis to _finite_ width linear networks, for one and more layers - under some assumptions, the authors show that the predictive variance of finite-width linear networks can be described largely by two noise sources: the _functional noise_ as described above, and the _kernel noise_ stemming from the variance over different kernels in the finite-width regime. Finally, the authors provide evidence that the discrepancy between a linearly trained model and it's non-linear, gradient descent counterpart, can be bounded on the order of it's layer-width, justifying the approximation of non-linear ensembles with linear models.

The second section (section 3) validates the authors hypothesis via a set of empirical experiments. The first set of experiments utilizes tractable linear models for which distinct components of predictive variance can be explicitly removed. Here the authors use a  toy dataset to provide an intuitive visualization of how each isolated component of the variance behaves. Next the authors study 2 real-world tasks, CIFAR-10 and MNIST, using a more complex linear model (both a CNN and MLP model), examining both ensemble performance and ensemble OOD detection rates. They find that different noise components have different levels of OOD detection - in some cases, the model with only kernel noise can achieve comparable or better performance in OOD detection than the full model. They then extend their empirical analysis to non-linear models - using a clever trick around model initialization, the authors derive two variants of gradient-descent models which isolate either functional initialization noise or initial kernel noise. They find that the isolated components of predictive variance in neural ensembles trained with (S)GD mirror those of the linearly trained models on OOD detection performance. Finally, the authors train fully modern neural networks, using residual connections, batch-norm, and cross-entropy loss on CIFAR-10, leveraging the same techniques to isolate each noise source. They show that, in this setting, isolating noise sources can have a significant effect on OOD performance - in particular, _centered_ models which remove initialization noise, outperform standard models.

**Questions:**

- The experiment in Figure 1 is a nice visualization, but the intuition it provides is not tied into later results - why is the distance or angle between datapoints a useful interpretation of functional or kernel noise? If it is not relevant to the next set of experiments due to data dependency then what is the toy experiment showing us?

**Limitations:**

yes

**Strengths And Weaknesses:**

Strengths:
- The predictive variance of neural networks, and it's utility to OOD detection is of broad interest to the field.
- Leveraging linear networks (and their kernels) to get a theoretical intuition about the different sources of variance in neural ensembles is a clever and novel approach to understanding the sources of variance, and their effects.
- The theoretical analysis provided in this work is very interesting, and provides the foundation for future work aiming to understand predictive variance in neural models.
- The final empirical result is relatively strong - namely, they show that a particular source of noise (kernel noise) is more effective at identifying OOD examples than standard neural models in modern neural architectures trained with SGD.

Weaknesses:
- The biggest weakness of this work lies in the empirical analysis and the takeaways from this analysis:
	- Overall, the benefits of each isolated noise source are not consistent. It is not clear what effect of each noise source has on OOD detection - in some cases isolated kernel noise outperforms isolated functional noise, and in other cases the opposite occurs.
	- The "big takeaway" of the role of each noise source in OOD detection largely seems to rely on the single result of the fully fleshed out neural network on CIFAR-10. Only is this model is there a clear story on the importance of isolated kernel noise.
	- In other words, the paper cannot empirically show us why the proposed decomposition is practically useful other than that it has _some effect_ on OOD detection.
- Lastly, I think the structure of the paper could be improved:
	- For instance, much of the theoretical analysis lacks intuition around key quantities (e.g. the matrix Q is referenced all over the paper, and it would help the readability of the formulas to provide some intuition as to what this matrix represents).
	- Figures and tables are placed distant from their corresponding sections - additionally, sometime the corresponding figure or table is not referenced in the main text (e.g. Table 1 is not referenced in section 3.2.2).

---

> ### Author Response · Authors · 2022-08-02
> **Response to Reviewer rqPL**
>
>
> We thank the reviewer for carefully evaluating our work and we are pleased with the overall positive feedback. We particularly appreciated the accuracy of the reviewer’s summary of our paper. We will try to address the reviewer’s concerns below point-by-point and hope that we can lift some of the criticism. Of course, we are happy to discuss any additional points during the rebuttal period.
>
> **P1**: The biggest weakness of this work lies in the empirical analysis and the takeaways from this analysis: Overall, the benefits of each isolated noise source are not consistent. It is not clear what effect of each noise source has on OOD detection - in some cases isolated kernel noise outperforms isolated functional noise, and in other cases the opposite occurs. The "big takeaway" of the role of each noise source in OOD detection largely seems to rely on the single result of the fully fleshed out neural network on CIFAR-10. Only is this model is there a clear story on the importance of isolated kernel noise. In other words, the paper cannot empirically show us why the proposed decomposition is practically useful other than that it has some effect on OOD detection.
>
> **A1**:  Thank you for bringing up these important points.
> In general, OOD detection depends strongly on the inductive bias induced by the neural network architecture, how the model is trained and naturally on the datasets considered out- and in-distribution. Therefore, we cannot make precise claims about improved OOD detection without a proper definition of what is considered OOD data. We want to clarify that we do not suggest any of our disentangled models and their respective predictive variances lead *generally* to improved OOD performance - rather, we only argue that the different models considered in our work induce significantly different inductive biases leading to different OOD detection, and the insight we get into them by analyzing the terms translates roughly to nonlinear deep ensembles as well. We hope our additional OOD baselines detailed in the general response illustrates this point.
>
> Nevertheless, as suggested by the WRN experiment as well as the additional large-scale network results mentioned in the general response, these distinct inductive biases seem to be particularly relevant for image classification tasks. We believe this trend is surprising and of practical relevance.
>
> We will work carefully on providing an improved discussion of this important point in the manuscript in section 3 and the conclusion.
>
> **P2**:  Lastly, I think the structure of the paper could be improved:
> For instance, much of the theoretical analysis lacks intuition around key quantities (e.g. the matrix Q is referenced all over the paper, and it would help the readability of the formulas to provide some intuition as to what this matrix represents).
>
> **A2**: Thank you for raising this issue. We are happy for any suggestions to make our paper more comprehensible and we will work on improving this aspect in the final manuscript. In particular, we will clearly detail the intuition behind the key variance terms, by deriving how  $\mathbb{V}^{\text{a}}$ relates to the infinite width variance, as well as clarifying the description of  $\mathbb{V}^{\text{i}}$ by moving elements of Appendix A.3.1 to the main text.
>
> **P3**: Figures and tables are placed distant from their corresponding sections - additionally, sometime the corresponding figure or table is not referenced in the main text (e.g. Table 1 is not referenced in section 3.2.2).
>
> **A3**:  Thank you for spotting this. We also agree with the reviewer that the placement of the Figure and Tables can be improved. We will fix this in the final manuscript

---

> > ### Author Response · Authors · 2022-08-02
> > **Response to Reviewer rqPL  pt 2**
> >
> > **Q1**:  The experiment in Figure 1 is a nice visualization, but the intuition it provides is not tied into later results - why is the distance or angle between datapoints a useful interpretation of functional or kernel noise? If it is not relevant ta the next set of experiments due to data dependency then what is the toy experiment showing us?
> >
> > **A4**: Our motivation for showing Figure 1 and 7 is the following.
> >
> > 1. We want to provide an intuition of how drastic the different inductive biases induced by our disentangled models are. As described above, we will work carefully on improving our manuscript to make the motivations and take-always of our empirical section and paper in general clearer.
> >
> > 2. The figure validates the closed-form description of one hidden layer neural network predictive variance, namely the angle and distance description derived in the Appendix.
> >
> > We now complement this intuition with our new baseline experiment outlined in the general response, which further indicate the OOD performance of  $\mathbb{V}^{\text{a}}$ and  $\mathbb{V}^{\text{c}}$ relate to this inductive bias.
> >
> > We again thank the reviewer for the constructive feedback, and hope we adequately addressed the reviewer’s questions. We are happy to discuss any additional points during the rebuttal period.

---

> > > ### Comment · Reviewer_rqPL · 2022-08-08
> > > **Response to the Authors**
> > >
> > > Thank you for your clarifying comments.
> > >
> > > >Therefore, we cannot make precise claims about improved OOD detection without a proper definition of what is considered OOD data.
> > >
> > > Thank you for the clarification - certainly I agree. My only issue is perhaps with the claim in the introduction "We highlight the practical implications of our theory by proposing methods to isolate unfavourable noise sources, improving OOD detection in realistic settings." - If I were a practitioner looking to improve the OOD performance of my ensemble in realistic settings, I may read the introduction and believe that the work contains a method I can implement to improve my system. Instead, the work is proposing a method to isolate _potentially_ unfavorable noise sources, which may or may not improve OOD detection in realistic settings. This is a minor point but I believe is largely the source of my expectations in my original review.
> > >
> > > > As described above, we will work carefully on improving our manuscript to make the motivations and take-always of our empirical section and paper in general clearer.
> > >
> > > Thank you - to me this is the greatest weakness of this work, as it is otherwise is a solid paper with reasonable contributions. I believe that my original score of a 7 is an appropriate score for this work.

---

### Official Review · Reviewer_VVkd · 2022-07-22

**Rating:** 7
**Confidence:** 2
**Soundness:** 3 good
**Presentation:** 3 good
**Contribution:** 3 good

**Summary:**

This paper provides a first order approximation of the variance of a one-layer neural network. This approximation is composed of 5 terms:

- $\mathbb V^a$ is the finite-width equivalent of the predictive variance of an infinite width NTK, and depends on functional noise at initialization
- $\mathbb V^c$ is an approximation of the predictive variance of a linearly-trained network, and depends only on the kernel noise.
- $\mathbb V^i$ is the predictive variance of an ensemble with a NTK kernel and a data-specific prior
- $\mathbb V^{cor}$ and $\mathbb V^{res}$ are two terms that are negligible in empirical experiments.

Based on this decomposition, the authors define ensembles that depend only on one or another source of noise ($f^c$ and $f^a$), and show that isolating these sources of noise can improve OOD detection.

**Questions:**

- How would one extend the decomposition of Prop. 2.2 in the presence of additional sources of noise? Would an approach such as [1] be sufficient?
- How are the AUROC values computed? Are they computed based on the (pointwise) predictive variances of the different ensembles?
- Line 219: "Interestingly, $\mathbb V[f^{lin}(x)]$ depicts a superposition of the three isolated variances." Is this not a validation of the decomposition of Prop 2.2? Why is this surprising?
- In cases where we have $\mathbb V[f^{lin}] \le \mathbb V[f^{lin-c}] + \mathbb V[f^{lin-a}]$, is this due to $\mathbb V_i$? Does $\mathbb V_i$ factor into defining favorable/unfavorable sources of noise?
- Are there any difficulties in obtaining high-quality estimates of the different variance terms? Are the empirical estimates unbiased?


[1] Understanding Double Descent Requires A Fine-Grained Bias-Variance Decomposition, Adlam & Pennington, 2020

**Limitations:**

The authors discuss different limitations of their work, mostly based around assumptions made about the behavior of NNs (e.g., 2.3). These assumptions are discussed in detail, evaluated empirically, and put in the general context of this work.

In my mind, the two major limitations of this work are the following:
- The interplay between the two sources of noise (initialization & kernel noise) is not analyzed empirically, despite appearing with equal importance in the decomposition of Prop. 2.2.
- I found the conclusions of the empirical analysis to be somewhat unclear. I believe the discussion could be improved by the authors stating early on which hypothesis they seek to verify, and how that hypothesis is or isn't validated by the experimental results. In the paper's current state, I find the conclusions to be somewhat unclear (which source of noise is conducive to OOD detection? Is a single source of noise sufficient in general?).

**Strengths And Weaknesses:**

# Strengths
- Understanding the degree to which the variance of a deep ensemble is indicative of an ensemble's certainty in making a prediction is a crucial question as deep ensembles are deployed on real-world applications.
- The authors push through a detailed decomposition of the variance of linearly trained neural networks, then generalize this decomposition under several assumptions to neural networks trained with gradient descent and SGD.
- The authors analyze the terms in the proposed decomposition empirically across a variety of experimental settings (different training modalities, datasets, and dataset samples).

# Weaknesses
- I found the empirical section of this paper difficult to put into context with the previous decomposition (cf. "Questions" section below).
- I am not sure I follow the author's claim that isolating unfavorable noise improves OOD detection.  This appears to be the case in certain experiments but not all (e.g., Table 1 for the CNN on Cifar-10), and it also seems like the "unfavorable" noise depends on the experiment setting (Table 1, linear models on CNN: $f^{lin-c}$ does better for SVHN but worse for LSUN).
- I would've liked to see $\mathbb V^i$ included in the empirical analysis, since it is the third major linear term in the approximation to the full variance.

---

> ### Author Response · Authors · 2022-08-02
> **Response to Reviewer VVkd**
>
>
> We thank the reviewer for carefully evaluating our work and the constructive comments on the paper. We will address the concerns one by one and hope that the additional clarification and the provided analyses will lead the reviewer to vote for an acceptance of the paper. We are eager to provide any additional information upon the request of the reviewer.
>
> **P1**: I am not sure I follow the author's claim that isolating unfavorable noise improves OOD detection. This appears to be the case in certain experiments but not all (e.g., Table 1 for the CNN on Cifar-10), and it also seems like the "unfavorable" noise depends on the experiment setting (Table 1, linear models on CNN: flin−c  does better for SVHN but worse for LSUN).
>
> **A1**:  Thank you for this important comment. As we will clarify in the main text, we do not claim that the proposed disentanglement leads to improved out-of-distribution detection *in general*. Instead, our main objective is to (1) provide a finer description of the predictive distribution of a deep ensemble, by decomposing the total variance of linear deep ensembles into terms that probably survive nonlinear training, and (2) empirically demonstrate the approximation holds in practice by verifying in particular that the OOD detection performance translates robustly from linearized to non-linearly trained models as shown in Table 1, for the variance terms that can be isolated. Finally, we complement our result by demonstrating that isolating specific noise affects the final performance even in a fully trained modern neural network, indicating that the identified and analyzed noises do in fact affect training in realistic settings, and each induce a specific inductive bias that can be favorable or not for the considered task.
>
> It is indeed rather surprising that, as observed by the reviewer, the disentangled function often (but not always) leads to better performance on common OOD datasets (see also our new results presented in the general response). This is not obvious as the standard ensemble could have more favorable inductive bias on the OOD datasets considered. Generally, OOD detection depends strongly on the inductive bias induced by the neural network architecture, how the model is trained and naturally on the datasets considered out- and in-distribution. Therefore, one can not make precise claims about improved OOD detection without a proper definition of what is considered OOD data, see for example for a discussion [1]. Therefore, we only argue that the different models considered in our work induce significantly different inductive biases leading to different (but not better) OOD detection. We hope our additional OOD baselines detailed in the general response illustrates this point.
>
> We will work carefully on providing an improved discussion of this important point in the manuscript in section 3 and the conclusion. Furthermore, we will aim to clarify this already in the introduction, as suggested by the reviewer, where we will stress these points and describe more clearly the aim of the paper. We hope that this will clarify the paper in general, but especially the motivation behind our empirical investigation and the conclusions one can draw from it.
>
> **P2**: I would've liked to see Vi included in the empirical analysis, since it is the third major linear term in the approximation to the full variance.
>
> **A2**: Thank you again for this great point and important comment.
> In order to add $\mathbb{V}^i$ and the corresponding OOD detection performance of the corresponding disentangled model in Table 1, we must compute the NTK inverse which is challenging for larger datasets as it requires inverting large matrices. While this is still possible for N=1000, it quickly becomes intractable in realistic settings, even for the full MNIST dataset. In order to show the correspondence of the predictive variance of linearized vs non-linear deep ensemble, we thus had decided to focus on terms that can easily be computed in the general setting, namely  $\mathbb{V}^a$ and  $\mathbb{V}^c$, which can be obtained using the trick we outline in section 3.2.2.
> Nevertheless, we agree with the reviewer that demonstrating the “inductive bias” of each identified variance term is also of great interest, regardless of the practical feasibility to exploit it. We are therefore computing the corresponding numbers now for CIFAR10 (N=1000) and will add them to the final manuscript, see numbers for MNIST (N=1000) in the general response.

---

> > ### Author Response · Authors · 2022-08-02
> > **Response to Reviewer VVkd pt 2**
> >
> > **Q1**: How would one extend the decomposition of Prop. 2.2 in the presence of additional sources of noise? Would an approach such as [1] be sufficient?
> >
> > **A3**: Thank you for pointing out this related work. To the best of our knowledge, it is not known how to describe the training of linearized / wide neural networks through the neural tangent kernel under stochastic gradient descent - which potentially could allow us to integrate this particular noise source into our description. Note however that we found that our description is robust towards stochastic gradient descent, see Table 5 in the appendix, and again the robust translation from linearized to gradient descent models (trained with large batches) for larger dataset size $N$ shown in Table 1. Please come back to us if you want to draw our attention to a particular method used in the work the reviewer cited that is relevant to our study.
> >
> > **Q2**: How are the AUROC values computed? Are they computed based on the (pointwise) predictive variances of the different ensembles?
> >
> > **A4**: Yes, we follow related work on this and compute the AUROC values based on the predictive variance computed on single out- and in-distribution data. We will add a description, including pseudocode, of how exactly these values are computed in the appendix.
> >
> > **Q3**: Line 219: "Interestingly, V[flin(x)] depicts a superposition of the three isolated variances." Is this not a validation of the decomposition of Prop 2.2? Why is this surprising?
> >
> > **A5**:  Thank you for spotting this misnomer. Although the decomposition is clearly described in Prop 2.2, the different variance terms usually have notably different magnitudes (see also Table 1) and therefore the overall influence on the total variance is not obvious. We will clarify this point.
> >
> > **Q4**: In cases where we have  V[flin]≤V[flin−c]+V[flin−a] , is this due to Vi ? Does Vi factor into defining favorable/unfavorable sources of noise?
> >
> > **A6**:  Thank you for bringing up this important point. $\mathbb{V}^{\text{lin-i}}$ is the variance of an actual random variable, and therefore is positive and will add to the full variance. In general, the total variance as the sum of the other terms only holds when many of our assumptions are satisfied, namely that the models are fully trained (which in practice is impossible to realize) or the validity of our Taylor approximation. In this particular case, we believe the discrepancy mainly stems from $\mathbb{V}^{\text{corr}}$  which is a covariance term, and can thus be negative. The covariance is usually small in the setups we considered (as shown in Appendix Figure 4), yet it could be of similar order of magnitude as the other terms in more realistic settings. This term was omitted for the same reason as $\mathbb{V}^{\text{i}}$ - its lack of practical interest due to the intractability of its computation in more practical settings. However, again, for the sake of completeness and clarity, we will add $\mathbb{V}^{\text{corr}}$ to the scaling plot in Figure 2 as well as Table 1. We thank the reviewer for raising this issue, which we believe overall makes the paper more complete.
> >
> > **Q5**: Are there any difficulties in obtaining high-quality estimates of the different variance terms? Are the empirical estimates unbiased?
> >
> > **A7**:   Yes, the estimates are unbiased. We will add analyses in the appendix where we show how the ensemble size i.e. the number of samples influences the variance estimates. Usually, we found an ensemble size of 5-10 members sufficient to obtain a stable estimate.
> >
> > **P3**: The interplay between the two sources of noise (initialization & kernel noise) is not analyzed empirically, despite appearing with equal importance in the decomposition of Prop. 2.2.
> >
> >  **A8**: We hope that our clarification of why these terms were omitted addresses this limitation. We will nevertheless add these quantities to Table 1 for N=1000 as we agree the demonstration of their inductive bias is of interest. We are of course happy to provide any additional information.

---

> > > ### Author Response · Authors · 2022-08-02
> > > **Response to Reviewer VVkd pt 3**
> > >
> > > **P4**: I found the conclusions of the empirical analysis to be somewhat unclear. I believe the discussion could be improved by the authors stating early on which hypothesis they seek to verify, and how that hypothesis is or isn't validated by the experimental results. In the paper's current state, I find the conclusions to be somewhat unclear (which source of noise is conducive to OOD detection? Is a single source of noise sufficient in general?).
> > >
> > >  **A9**: We hope again that our discussion above as well as the general response addresses these limitations.
> > >
> > > We thank the reviewer again for the very constructive feedback. We hope the additional analyses of the correlation terms will provide additional clarifying insight into their inductive biases. Most importantly, we thank the reviewer for pointing out the lack of clarity in describing the aim of the paper as well as the purpose of the empirical section. We hope that given the additional clarification, together with the renewed strong empirical evidence provided in the general response,  the reviewer will consider voting for acceptance of the paper
> > >
> > > [1] Are Bayesian neural networks intrinsically good at out-of-distribution detection? Henning et al, 2021

---

> > > > ### Comment · Reviewer_VVkd · 2022-08-09
> > > > **Rebuttal response**
> > > >
> > > > I thank the authors for their in-depth reply to the questions I raised; their response has addressed the concerns I had and clarified any remaining misunderstandings.
> > > >
> > > > I will adjust my score accordingly; thanks again for the extensive discussion in a short amount of time.

---

> ### Author Response · Authors · 2022-08-09
> **Happy to provide final clarifications**
>
> Dear reviewer VVkd,
>
> Many thanks again for your helpful comments and the effort in evaluating our work carefully. We want to politely remind you that the author-reviewer discussion period is shortly coming to an end. Please do not hesitate to contact us before the interactive discussion period ends to discuss any further concerns.

---

### Author Response · Authors · 2022-08-02
**General response**

We thank all the reviewers for their detailed feedback as well as their constructive and overall encouraging comments on the paper. We worked intensively on addressing the reviewer's criticism and suggestions which can be found in the detailed individual responses. Here, we want to draw attention to the following new results which we believe strengthen the paper in several aspects.

**Additional empirical results on large scale experiments**

Based on the general interest in our large-scale experiments, we extend our results of Table 2 by

1. training a WRN-28-10 on CIFAR100 and measuring AUROC OOD performance on the same datasets as for CIFAR10.
2. training an AlexNet on FashionMNIST and measuring AUROC OOD performance on MNIST, EMNIST and KMNIST.
3. training a WRN-28-10 on CIFAR10 and CIFAR100 with the MSE loss instead of cross-entropy. This setup is motivated by our theory as it requires the MSE loss to hold and indeed shows greater effect of the initial noise on the final predictive variance.

Furthermore we now indicate statistically significant improvement upon the baseline ( $f^{\text{sgd}}$ ) in bold, with 5 seeds and p>0.8 (re reviewer e4Rz).

Overall, we confirm that the initial noise does indeed affect the predictive variance of fully trained models and therefore affect their OOD detection performance significantly, even when diverging from toy settings. We also note the surprising performance of $f^{\text{sgd-c}}$ on many of these tasks. Please see the individual responses and **Changes to the manuscript** below where we address the motivation and presentation of these results in more detail.

| Training setup       | Model              | Test (%)          | Other CIFAR           | SVHN                  | LSUN                  | TIN                   | iSUN                  |
|----------------------|--------------------|-------------------|-----------------------|-----------------------|-----------------------|-----------------------|-----------------------|
| CIFAR10  - CE Loss   |   $f^{\text{sgd}}$ | 89.36$^{\pm .36}$ |     .829$^{\pm .001}$ |     .900$^{\pm .002}$ | .891$^{\pm .002}$     | .860$^{\pm .001}$     | .883$^{\pm .001}$     |
|                      | $f^{\text{sgd-a}}$ | 89.01$^{\pm .32}$ |     .827$^{\pm .002}$ |     .894$^{\pm .003}$ | .887$^{\pm .003}$     | .855$^{\pm .004}$     | .880$^{\pm .002}$     |
|                      | $f^{\text{sgd-c}}$ | 89.56$^{\pm .30}$ | **.831$^{\pm .003}$** |     .900$^{\pm .004}$ | **.895$^{\pm .003}$** | **.863$^{\pm .003}$** | **.885$^{\pm .003}$** |
|                      |                    |                   |                       |                       |                       |                       |                       |
| CIFAR10  - MSE Loss  | $f^{\text{sgd}}$   | 77.94$^{\pm .22}$ |     .739$^{\pm .004}$ |     .863$^{\pm .006}$ | .823$^{\pm .007}$     | .795$^{\pm .007}$     | .813$^{\pm .008}$     |
|                      | $f^{\text{sgd-a}}$ | 75.34$^{\pm .18}$ |     .707$^{\pm .004}$ |     .841$^{\pm .011}$ | .784$^{\pm .003}$     | .763$^{\pm .010}$     | .761$^{\pm .002}$     |
|                      | $f^{\text{sgd-c}}$ | 77.88$^{\pm .30}$ |     .739$^{\pm .001}$ | **.880$^{\pm .006}$** | **.830$^{\pm .005}$** | **.807$^{\pm .006}$** | .813$^{\pm .005}$     |
|                      |                    |                   |                       |                       |                       |                       |                       |
| CIFAR100  - CE Loss  | $f^{\text{sgd}}$   | 67.57$^{\pm .37}$ |     .703$^{\pm .003}$ |     .776$^{\pm .005}$ | .735$^{\pm .003}$     | .742$^{\pm .004}$     | .741$^{\pm .003}$     |
|                      | $f^{\text{sgd-a}}$ | 67.26$^{\pm .11}$ |     .705$^{\pm .003}$ |     .773$^{\pm .003}$ | .735$^{\pm .003}$     | .741$^{\pm .001}$     | .742$^{\pm .002}$     |
|                      | $f^{\text{sgd-c}}$ | 67.59$^{\pm .35}$ | **.708$^{\pm .003}$** |     .778$^{\pm .004}$ | **.738$^{\pm .004}$** | .744$^{\pm .005}$     | .744$^{\pm .006}$     |
|                      |                    |                   |                       |                       |                       |                       |                       |
| CIFAR100  - MSE Loss | $f^{\text{sgd}}$   | 63.00$^{\pm .15}$ |     .704$^{\pm .003}$ |     .741$^{\pm .005}$ | .715$^{\pm .004}$     | .739$^{\pm .005}$     | .722$^{\pm .005}$     |
|                      | $f^{\text{sgd-a}}$ | 62.19$^{\pm .24}$ | **.710$^{\pm .004}$** |     .740$^{\pm .009}$ | **.729$^{\pm .006}$** | **.749$^{\pm .004}$** | **.730$^{\pm .004}$** |
|                      | $f^{\text{sgd-c}}$ | 62.90$^{\pm .06}$ |     .705$^{\pm .002}$ | **.746$^{\pm .004}$** | **.720$^{\pm .005}$** | **.743$^{\pm .004}$** | .725$^{\pm .006}$     |

---

> ### Author Response · Authors · 2022-08-02
> **General response pt 2**
>
>
>
> | Training setup         | Model              | Test (%)          | MNIST                  | EMNIST                 | KMNIST             |
> |------------------------|--------------------|-------------------|------------------------|------------------------|--------------------|
> | FashionMNIST - CE Loss | $f^{\text{sgd}}$   | 93.22$^{\pm .39}$ | 0.868$^{\pm .014}$     | 0.856$^{\pm .004}$     | 0.935$^{\pm .006}$ |
> |                        | $f^{\text{sgd-a}}$ | 93.12$^{\pm .09}$ | **0.880$^{\pm .01}$**  | 0.838$^{\pm .006}$     | 0.926$^{\pm .003}$ |
> |                        | $f^{\text{sgd-c}}$ | 93.21$^{\pm .13}$ | **0.883$^{\pm .007}$** | **0.867$^{\pm .011}$** | 0.933$^{\pm .005}$ |
>
>
> **Extention of the Table 1 analyses to  $f^{\text{lin-i}}$, $f^{\text{lin-cor}}$ and their variance**
>
>
> Based on the suggestions of Reviewer VVkd, we will extend the main papers' analyses by including $f^{\text{lin-i}}, \mathbb{V}[f^{\text{lin-i}}]$ in Table 1 for the moderate dataset size $N=1000$. While the practical interest of these quantities is limited as it cannot easily be isolated in the general setting, demonstrating the “inductive bias” of each identified variance term is also of great interest, regardless of the practical feasibility to exploit it. The table below presents the result for MNIST - CIFAR10 numbers are still being computed at the moment and will be added to the final manuscript.
>
> |                    | $\mathbb{V}[f(x)]$ | Test (%)     | FM           | EM           | KM           |
> |--------------------|------------------|--------------|--------------|--------------|--------------|
> | $f^{\text{lin-i}}$ |    0.225$^{\pm .004}$ | 89.6 $^{\pm .17}$  | .980$^{\pm .001}$ | .895$^{\pm .003}$ | .990$^{\pm .002}$ |
>
> **Simple AUROC baseline illustrating the inductive biases**
>
> To illustrate the inductive biases of each of $V^a$ and $V^c$ elucidated in the study of single hidden layer ReLU networks, we add a simple baseline experiment in the Appendix.
>
> $V^a$ and $V^c$ loosely measure OOD with respectively an L2-distance and angle-based metric, as can be visualized in Figure 1 and Figure 7 described analytical for single hidden layer ReLU network. Inspired by the comments from reviewer GBTo, we illustrate these inductive biases by drawing a correspondence of the OOD abilities of  $V^a$ and $V^c$ with those of simple kernel models: an RBF kernel and a cosine-similarity kernel, measured in the input space. As can be seen in the table below, their relative performance at OOD detection between (1) CIFAR10 and SVHN, LSUN as well as iSUN and (2) MNIST and FM, EM as well as KM, mirrors those of  $V^a$ and $V^c$ well, even for deeper MLP and Conv networks.
>
> | Method | C10-SVHN | C10-LSUN | C10-iSUN | MNIST-FM | MNIST-EM | MNIST-KM |
> |--------|----------|----------|----------|----------|----------|----------|
> | Angle  | .668     | .648     | .684     | .980     | .916     | .995     |
> | RBF    | .580     | .756     | .750     | .991     | .923     | .992     |
>
> This indicates that the inductive biases induced by these deeper and even convolutional models seem to follow the analytic description of the one-hidden layer MLP case. We believe that this additional analysis gives a nice intuition of what influences the variance magnitude and OOD detection of the disentangled models analyzed in the paper.
>
> **Changes to the manuscript**
>
> Based on the suggestions, we will improve the presentation of our theoretical and empirical results.
>
> 1. We want to stress that we do not claim any of the noises are unfavorable in general. Rather, our main takeaway from the OOD detection results (re reviewer rpPl, CZjD), is that the analyzed variance terms each provide a distinct inductive bias, and that isolating them *at initialization* leads to a significant change in the resulting *fully trained* function, predictive variance and therefore also OOD detection. Nevertheless, which of these functions performs better on which OOD dataset cannot be said, neither by our method nor by any other method, without a proper definition of OOD data. See e.g. higher AUROC values for  $f^{\text{sgd-a}}$ compared to $f^{\text{sgd-c}}$ on CIFAR100, whereas the opposite is true for CIFAR10. We will stress this important point and takeaway in the final manuscript.
> 2. Based on the concerns of reviewer e4Rz, we will extend our discussion (lines 131-137) of the assumptions of our variance description and its implication.
> 3. We will provide a detailed description, including  pseudocode, of how we compute the AUROC and variance numbers in the paper (re reviewer VVkd, GBTo).
> 4. Based on the suggestions of reviewer e4Rz we will make propositions 2.1 and 2.2. more concise by stating the needed assumptions more clearly.
>
> We again thank all the reviewers for their help in highlighting the shortcomings of our paper, and hope the suggested changes and provided clarifications will convince them to raise the score.

---

### Author Response · Authors · 2022-08-08
**Happy to provide final clarifications**

Thank you all once again for your useful comments. The end of the author-reviewer discussion period is coming close, and we see that some of you have not yet updated your reviews in response to our new results and proposed changes addressing your concerns. In case something is unclear, please do not hesitate in contacting us before the interactive discussion period ends.

---

### Meta-Review · Area_Chair_F3ep · 2022-08-29

**Recommendation:** Accept
**Confidence:** Certain

**Metareview:**

The paper proposes to decompose the predictive variance of deep ensembles, into different sources of noise, by studying linearly trained finite width neural networks.

The reviewers found the paper well-written, the theoretical analysis interesting, and the experimental evaluation adequate to justify the key claims.

I also appreciate the detailed author rebuttal to the reviewers’ questions. During the discussion, all the reviewers leaned towards acceptance. I encourage authors to address remaining comments in the final manuscript.

I recommend accept. Nice work!

**Award:**

No

---

### Decision · Program_Chairs · 2022-09-14

Accept